



# Sensitivity of ice sheet surface velocity and elevation to variations in basal friction and topography in the Full Stokes and Shallow Shelf Approximation frameworks

Gong Cheng[1], Nina Kirchner[2,3], and Per Lötstedt[1]

[1]Department of Information Technology, Uppsala University, P. O. Box 337, SE-751 05 Uppsala, Sweden
[2]Department of Physical Geography, Stockholm University, SE-106 91 Stockholm, Sweden
[3]Bolin Centre for Climate Research, Stockholm University, SE-106 91 Stockholm, Sweden

**Correspondence:** Gong Cheng (cheng.gong@it.uu.se)

**Abstract.** Predictions of future mass loss from ice sheets are afflicted with uncertainty, caused, among others, by insufficient understanding of spatio-temporally variable processes at the inaccessible base of ice sheets for which few direct observations exist and of which basal friction is a prime example. Here, we use an inverse modeling approach and the associated time-dependent adjoint equations, derived in the framework of a Full Stokes model and a Shallow Shelf/Shelfy Stream Approximation model, respectively, to determine the sensitivity of ice sheet surface velocities and elevation to perturbations in basal friction and basal topography. Analytical and numerical examples are presented showing the importance of including the time dependent kinematic free surface equation for the elevation and its adjoint, in particular for observations of the elevation. A closed form of the analytical solutions to the adjoint equations is given for a two dimensional vertical ice in steady state under the Shallow Shelf Approximation. There is a delay in time between a perturbation at the ice base and the observation of the change in elevation. A perturbation at the base in the topography has a direct effect in space at the surface above the perturbation and a perturbation in the friction is propagated directly to the surface in time. Perturbations with long wavelength and low frequency will propagate to the surface while those of short wavelength and high frequency are damped.

## 1 Introduction

Over the last decades, ice sheets and glaciers have experienced mass loss due to global warming, both in the polar regions, but also outside of Greenland and Antarctica (Farinotti et al., 2015; Mouginot et al., 2019; Pörtner et al., 2019; Rignot et al., 2019). The most common benchmark date for which future ice sheet and glacier mass loss and associated global mean sea level rise is predicted is the year 2100 AD, but recently, even the year 2300 AD and beyond are considered (Pörtner et al., 2019; Steffen et al., 2018). Global mean sea level rise is predicted to continue well beyond 2100 AD in the warming scenarios typically referred to as RCPs' (Representative Concentration Pathways, see van Vuuren et al. (2011)), but rates and ranges are afflicted with uncertainty, caused by, among others, insufficient understanding of spatio-temporally variable processes at the inaccessible base of ice sheets and glaciers (Pörtner et al., 2019; Ritz et al., 2015). These include the geothermal heat regime, subglacial and base-proximal englacial hydrology, and particularly, the sliding of ice sheet and glaciers across their





base, for which only few direct observations exist (Fisher et al., 2015; Key and Siegfried, 2017; Maier et al., 2019; Pattyn and Morlighem, 2020).

In computational models of ice dynamics, the description of sliding processes, including their parametrization, plays a central role, and can be treated in two fundamentally different ways, viz. using a so-called forward approach on the one hand, or an

inverse approach on the other hand. In a forward approach, an equation referred to as a sliding law is derived from a conceptual friction model, and provides a boundary condition to the equations describing the dynamics of ice flow (in glaciology often referred to as the Full Stokes (FS) model) and which, once solved, render e.g. ice velocities as part of the solution. Studies of frictional models and resulting sliding laws for glacier and ice sheet flow emerged in the 1950s, see e.g. Fowler (2011); Iken (1981); Lliboutry (1968); Nye (1969); Schoof (2005); Weertman (1957), and have subsequently been implemented into

numerical models of ice sheet and glacier behavior e.g. in Brondex et al. (2017, 2019); Gladstone et al. (2017); Tsai et al. (2015); Wilkens et al. (2015); Yu et al. (2018), and continue to be discussed (Zoet and Iverson, 2020), occasionally controversially (Minchew et al., 2019; Stearn and van der Veen, 2018).

Because little or no observational data is available to constrain the parameters in such sliding laws (Minchew et al., 2016; Sergienko and Hindmarsh, 2013), actual values of the former, and their variation over time (Jay-Allemand et al., 2011; Schoof,

2010; Sole et al., 2011; Vallot et al., 2017), often remain elusive. Yet, they can be obtained computationally by solving an inverse problem provided that observations of e.g. ice velocities at the ice surface, and elevation of the ice surface, are available (Gillet-Chaulet et al., 2016; Isaac et al., 2015). Note that the same approach, here described for the case of the sliding law, can be used to determine other "inaccessibles", such as optimal initial conditions for ice sheet modeling (Perego et al., 2014), the sensitivity of melt rates beneath ice shelves in response to ocean circulation (Heimbach and Losch, 2012), or to estimate basal

topography beneath an ice sheet (van Pelt et al., 2013). The latter is not only difficult to separate from the determination of the sliding properties (Kyrke-Smith et al., 2018; Thorsteinsson et al., 2003), but also has limitations related to the spatial resolution of surface data and/or measurement errors, see Gudmundsson (2003, 2008); Gudmundsson and Raymond (2008).

Adopting an inverse approach, the strategy is to minimize an objective function describing the deviation of observed target quantities (such as the ice velocity) from their counterparts as predicted following a forward approach when a selected param-

eter in the forward model (such as the friction parameter in the sliding law) is varied. The gradient of the objective function is computed by solving the so-called adjoint equations to the forward equations, where the latter often are slightly simplified, such as e.g. by assuming a constant ice thickness or a constant viscosity (MacAyeal, 1993; Petra et al., 2012). However, when inferring friction parameter(s) in a sliding law using an inverse approach, recent work (Goldberg et al., 2015; Jay-Allemand et al., 2011) has shown that it is not sufficient to consider the time-independent (steady state) adjoint to the momentum balance

in the FS model. Rather, it is necessary to include the time-dependent advection equation for the ice surface elevation in the inversion. Likewise, but perhaps more intuitively understandable, the choice of the underlying glaciological model (FS model, vs. e.g. Shallow shelf/stream approximation (SSA) model, see Sect. 2), has an impact on the values of the friction parameters obtained from the solution of the corresponding inverse problem (Gudmundsson, 2008; Schannwell et al., 2019).

Here, we present an analysis of the sensitivity of the velocity field and the elevation of the surface of a dynamic ice sheet

(modelled by both FS and SSA, respectively, briefly described in Sect. 2) to perturbations in the sliding parameters contained





in Weertman's law (Weertman, 1957) and the topography at the ice base. The adjoint problem that is solved here to determine this sensitivity (Sect. 3) goes beyond similar earlier works by MacAyeal (1993); Petra et al. (2012) because it includes the time-dependent advection equation for the kinematic free surface. The key concepts and steps introduced in Sect. 3 are supplemented by detailed derivations, collected in Appendix A. The same adjoint equations are applicable in the inverse problem to compute

the gradient of the objective function and to quantify the uncertainty in the surface velocity and elevation due to uncertainties at the ice base. Analytical solutions in two dimensions of the stationary adjoint equations subjected to simplifying assumptions are presented, from which the dependence of the parameters, on e.g. friction coefficients and bedrock topography, becomes obvious. The time dependent adjoint equations are solved numerically, and the sensitivity to perturbations varying in time is investigated and illustrated. The accuracy of the analytical solutions and the adjoint approach has been discussed in a

companion paper (Cheng and Lötstedt, 2020), supported by extensive numerical computations.

## 2    Ice models

In this section, the equations emerging from adopting a forward approach of describing ice dynamics are presented, together with relevant boundary conditions, for the FS (4) and SSA model (8), respectively. These, and the notation and terminology introduced here, provide the framework in which the adjoint equations are discussed in Sect. 3.

The flow of large bodies of ice is described with the help of the conservation laws of mass, momentum and energy (Greve and Blatter, 2009), which together pose a system of non-linear partial differential equations (PDEs) commonly referred to as the FS equations in glaciological applications. In the FS equations, nonlinearity is introduced through the viscosity in Glen's flow law, a constitutive relation between strain rates and stresses (Glen, 1955). Continental sized ice masses (ice sheets, and, if applicable, their floating extensions known as ice shelves), are shallow in the sense that their vertical extension $V$ is orders

of magnitudes smaller than their horizontal extension $L$, such that the aspect ratio $V/L$ is in the order $10^{-2} - 10^{-3}$. The aspect ratio is used to introduce simplifications to the FS equations, resulting e.g. in the Shallow Ice (SIA) (Hutter, 1983), Shallow Shelf (Morland, 1987), and Shelfy Stream (MacAyeal, 1989) Approximations, parts of which can be coupled to FS using the ISCAL framework (Ahlkrona et al., 2016). They are all characterized by substantially reduced computational costs for numerical simulation, compared to using the FS model. A common simplification, also adopted in our analysis in the

following, is the assumption of isothermal conditions, which implies that the balance of energy need not be considered.
The upper surface of the ice mass, and also the ice-ocean interface, constitute a moving boundary and satisfy an advection equation describing the evolution of its elevation and location (in response to mass gain, mass loss, or/and mass advection). For ice masses resting on bedrock or sediments, sliding needs to be parameterized at the interface. The interface between floating ice shelves and sea water in the ice shelf cavities is usually regarded as frictionless.

## 2.1    Full Stokes model

We adopt standard notation and denote vectors and matrices in three-dimensional space by bold characters, and derivatives with respect to the spatial coordinates and time by subscripts $x, y, z$ and $t$. The horizontal plane $\omega$ is spanned by the $x$ and



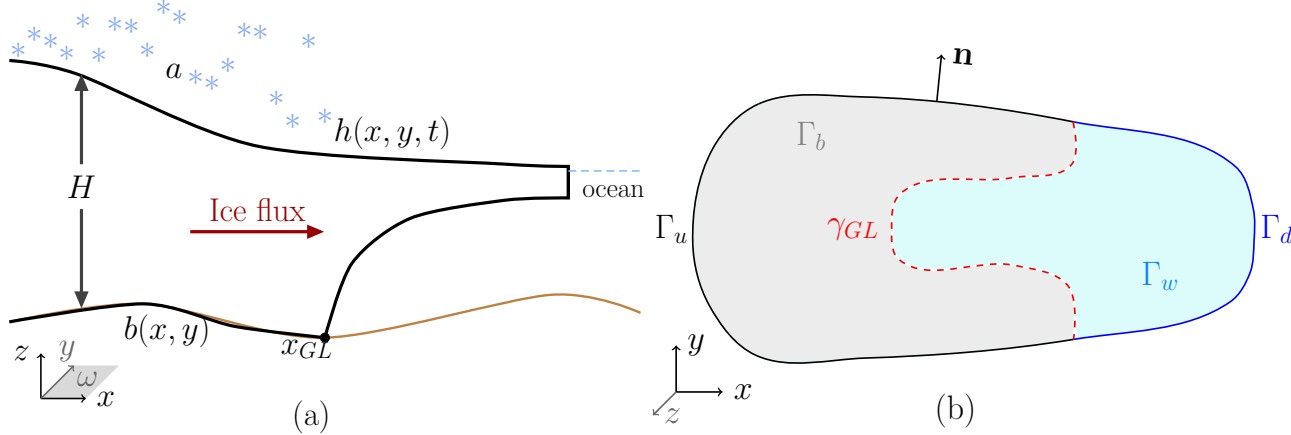

**Figure 1.** A schematic view of an ice sheet in the (a) $x - z$ plane and (b) $x - y$ plane.

$y$ coordinates, and $z$ is the coordinate in the vertical direction, see Fig. 1(a). Specifically, we denote by $u_1, u_2$, and $u_3$ the velocity components of $\boldsymbol{u} = (u_1, u_2, u_3)^T$ in the $x, y$ and $z$ direction, where $\boldsymbol{x} = (x, y, z)^T$ is the position vector and $T$ denotes the transpose. Further, the elevation of the upper ice surface is denoted by $h(x, y, t)$, the elevation of the bedrock and the location of the base of the ice are $b(x, y)$ and $z_b(x, y, t)$, and the ice thickness is $H = h - z_b$. Upstream of the grounding line,

$\gamma_{GL}, z_b = b$ and downstream of $\gamma_{GL}$ we have $z_b > b$, see Fig. 1. In two dimensions, $\gamma_{GL}$ consists of one point with $x$-coordinate $x_{GL}$.

The boundary $\Gamma$ enclosing the domain $\Omega$ occupied by the ice has different parts, see Fig. 1(b): The lower boundaries in the $\omega$ plane are denoted by $\Gamma_b$ (where the ice is grounded at bedrock), and $\Gamma_w$ (where the ice has lifted from the bedrock and is floating on the ocean). These two regions are separated by the grounding line $\gamma_{GL}$, defined by $(x_{GL}(y), y)$ based on the

10 assumption that ice flow is mainly along the $x$-axis. The upper boundary in the $\omega$ plane is denoted by $\Gamma_s$ (ice surface) at $h(x, y, t)$ in Fig. 1(a). The boundary of $\omega$ itself is denoted by $\gamma$.

The vertical lateral boundary (in the $x - z$ plane, Fig. 1(b)) has an upstream part denoted by $\Gamma_u$ in black and a downstream part denoted by $\Gamma_d$ in blue, where $\Gamma = \Gamma_u \cup \Gamma_d$. Obviously, if $\boldsymbol{x} \in \Gamma_u$, then $(x, y) \in \gamma_u$ or if $\boldsymbol{x} \in \Gamma_d$, then $(x, y) \in \gamma_d$ where $\gamma = \gamma_u \cup \gamma_d$. Letting $\boldsymbol{n}$ be the outward pointing normal on $\Gamma$ (or $\gamma$ in two dimensions $(x, y)$), the nature of ice flow renders the

15 conditions $\boldsymbol{n} \cdot \boldsymbol{u} \leq 0$ at $\Gamma_u$ and $\boldsymbol{n} \cdot \boldsymbol{u} > 0$ at $\Gamma_d$. In a two dimensional vertical ice (Fig. 1(a)), this corresponds to $\boldsymbol{x} = (x, z)^T$, $\omega = [0, L]$, $\gamma_u = 0$, and $\gamma_d = L$ where $L$ is the horizontal length of the domain. In summary, the domains are defined as:

$$
\begin{aligned}
\Omega &= \{\boldsymbol{x} | (x, y) \in \omega, z_b(x, y, t) \leq z \leq h(x, y, t)\}, \\
\Gamma_s &= \{\boldsymbol{x} | (x, y) \in \omega, z = h(x, y, t)\}, \\
\Gamma_b &= \{\boldsymbol{x} | (x, y) \in \omega, z = z_b(x, y, t), x < x_{GL}(y)\}, \\
\Gamma_w &= \{\boldsymbol{x} | (x, y) \in \omega, z = z_b(x, y, t), x > x_{GL}(y)\}, \\
\Gamma_u &= \{\boldsymbol{x} | (x, y) \in \gamma_u, z_b(x, y, t) \leq z \leq h(x, y, t)\}, \\
\Gamma_d &= \{\boldsymbol{x} | (x, y) \in \gamma_d, z_b(x, y, t) \leq z \leq h(x, y, t)\}.
\end{aligned}
\tag{1}
$$





Before the forward FS equations for the evolution of the ice surface $\Gamma_s$ and the ice velocity in $\Omega$ can be given, further notation needs to be introduced: ice density is denoted by $\rho$, accumulation and/or ablation rate on $\Gamma_s$ by $a$, and gravitational acceleration by $\boldsymbol{g}$. On $\Gamma_s$, $\boldsymbol{h} = (h_x, h_y, -1)^T$ describes the spatial gradient of the ice surface (in two vertical dimensions $\boldsymbol{h} = (h_x, -1)^T$). The strain rate $\mathbf{D}$ and the viscosity $\eta$ are given by

$$\mathbf{D}(\boldsymbol{u}) = \tfrac{1}{2}(\nabla\boldsymbol{u} + \nabla\boldsymbol{u}^T), \ \eta(\boldsymbol{u}) = \tfrac{1}{2}A^{-\frac{1}{n}}(\mathrm{tr}\mathbf{D}^2(\boldsymbol{u}))^\nu, \ \nu = \tfrac{1-n}{2n}, \tag{2}$$

where $\mathrm{tr}\mathbf{D}^2$ is the trace of $\mathbf{D}^2$. The rate factor $A$ in (2) depends on the temperature and Glen's flow law determines $n > 0$, here taken to be $n = 3$. The stress tensor is

$$\boldsymbol{\sigma}(\boldsymbol{u}, p) = 2\eta\mathbf{D}(\boldsymbol{u}) - \mathbf{I}p, \tag{3}$$

where $p$ is the isotropic pressure and $\mathbf{I}$ is the identity matrix.

Turning to the ice base, the basal stress on $\Gamma_b$ is related to the basal velocity using an empirical friction law. The friction coefficient has a general form $\beta(\boldsymbol{u}, \boldsymbol{x}, t) = C(\boldsymbol{x}, t)f(\boldsymbol{u})$ where $C(\boldsymbol{x}, t)$ is independent of the velocity $\boldsymbol{u}$ and $f(\boldsymbol{u})$ represents some linear or nonlinear function of $\boldsymbol{u}$. For instance, $f(\boldsymbol{u}) = \|\boldsymbol{u}\|^{m-1}$ with the norm $\|\boldsymbol{u}\| = (\boldsymbol{u} \cdot \boldsymbol{u})^{\frac{1}{2}}$ introduces a Weertman type friction law (Weertman, 1957) on $\omega$ with a Weertman friction coefficient $C(\boldsymbol{x}, t) > 0$ and an exponent parameter $m > 0$. Common choices of $m$ are $\frac{1}{3}$ and 1. Finally, a projection (Petra et al., 2012) on the tangential plane of $\Gamma_b$ is denoted by

$\mathbf{T} = \mathbf{I} - \boldsymbol{n} \otimes \boldsymbol{n}$ where the Kronecker outer product between two vectors $\boldsymbol{a}$ and $\boldsymbol{c}$ or two matrices $\mathbf{A}$ and $\mathbf{C}$ is defined by

$$(\boldsymbol{a} \otimes \boldsymbol{c})_{ij} = a_i c_j, \ (\mathbf{A} \otimes \mathbf{C})_{ijkl} = A_{ij}C_{kl}.$$

With these prerequisites at hand, the forward FS equations and the advection equation for the ice sheet's elevation and velocity for incompressible ice flow are

$$h_t + \boldsymbol{h} \cdot \boldsymbol{u} = a, \ \text{on } \Gamma_s, \quad t \geq 0,$$

$$h(\boldsymbol{x}, 0) = h_0(\boldsymbol{x}), \ \boldsymbol{x} \in \omega, \quad h(\boldsymbol{x}, t) = h_\gamma(\boldsymbol{x}, t), \ \boldsymbol{x} \in \gamma_u,$$

$$-\nabla \cdot \boldsymbol{\sigma}(\boldsymbol{u}, p) = -\nabla \cdot (2\eta(\boldsymbol{u})\mathbf{D}(\boldsymbol{u})) + \nabla p = \rho\boldsymbol{g}, \quad \nabla \cdot \boldsymbol{u} = 0, \ \text{in } \Omega(t), \tag{4}$$

$$\boldsymbol{\sigma}\boldsymbol{n} = \mathbf{0}, \ \text{on } \Gamma_s,$$

$$\mathbf{T}\boldsymbol{\sigma}\boldsymbol{n} = -Cf(\mathbf{T}\boldsymbol{u})\mathbf{T}\boldsymbol{u}, \quad \boldsymbol{n} \cdot \boldsymbol{u} = 0, \ \text{on } \Gamma_b,$$

$$\boldsymbol{\sigma}\boldsymbol{n} = -p_w\boldsymbol{n}, \ \text{on } \Gamma_w,$$

where $p_w$ is the water pressure at $\Gamma_w$, $h_0(\boldsymbol{x})$ is the initial surface elevation, and $h_\gamma(\boldsymbol{x}, t)$ is a given height on the inflow boundary. The boundary conditions for the velocity on $\Gamma_u$ and $\Gamma_d$ are of Dirichlet type such that

$$\boldsymbol{u}|_{\Gamma_u} = \boldsymbol{u}_u, \quad \boldsymbol{u}|_{\Gamma_d} = \boldsymbol{u}_d, \tag{5}$$

where $\boldsymbol{u}_u$ and $\boldsymbol{u}_d$ are known. These are general settings of the inflow and outflow boundaries which keep the formulation of the adjoint equations as simple as in Petra et al. (2014). Should $\Gamma_u$ be at the ice divide, the horizontal velocity is set to $\boldsymbol{u}|_{\Gamma_u} = \mathbf{0}$.

The vertical component of $\boldsymbol{\sigma}\boldsymbol{n}$ vanishes on $\Gamma_u$.





## 2.2   Shallow shelf approximation.

The three dimensional FS problem (4) in $\Omega$ can be simplified to a two dimensional, horizontal problem with $\boldsymbol{x} = (x, y) \in \omega$, by adopting the SSA, in which only $\boldsymbol{u} = (u_1, u_2)^T$ is considered. This is because the basal shear stress is negligibly small at the base of the floating part of the ice mass, viz. the ice shelf, rendering the horizontal velocity components almost constant in

the $z$ direction (Greve and Blatter, 2009; MacAyeal, 1989; Schoof, 2007). The SSA is often also used in regions of fast-flow over lubricated bedrock (MacAyeal, 1989; Pattyn et al., 2012).

The simplifications associated with adopting the SSA imply that the viscosity (see (2) for the FS model) is now given by

$$\eta(\boldsymbol{u}) = \frac{1}{2} A^{-\frac{1}{n}} \left( u_{1x}^2 + u_{2y}^2 + \frac{1}{4}(u_{1y} + u_{2x})^2 + u_{1x}u_{2y} \right)^{\nu} = \frac{1}{2} A^{-\frac{1}{n}} \left( \frac{1}{2} \mathbf{B} : \mathbf{D} \right)^{\nu}, \tag{6}$$

where $\mathbf{B}(\boldsymbol{u}) = \mathbf{D}(\boldsymbol{u}) + \nabla \cdot \boldsymbol{u}\, \mathbf{I}$ with $\nabla \cdot \boldsymbol{u} = \operatorname{tr} \mathbf{D}(\boldsymbol{u})$. This $\eta$ differs from (2) because $\mathbf{B} \neq \mathbf{D}$ due to the cryostatic approximation

of $p$ in the SSA. In (6), the Frobenius inner product between two matrices $\mathbf{A}$ and $\mathbf{C}$ is used, defined by $\mathbf{A} : \mathbf{C} = \sum_{ij} A_{ij} C_{ij}$. The vertically integrated stress tensor $\varsigma(\boldsymbol{u})$ (cf. (3) for the FS model) is given by

$$\varsigma(\boldsymbol{u}) = 2H\eta \mathbf{B}(\boldsymbol{u}). \tag{7}$$

The friction law in the SSA model is defined as in the FS case. Note that basal velocity is replaced by the horizontal velocity. This is possible because vertical variations in the horizontal velocity are neglected in SSA. Then, Weertman's law is

$\beta(\boldsymbol{u}, \boldsymbol{x}, t) = C(\boldsymbol{x}, t) f(\boldsymbol{u}) = C(\boldsymbol{x}, t) \|\boldsymbol{u}\|^{m-1}$ with a friction coefficient $C(\boldsymbol{x}, t) \geq 0$, just as in the FS model. In summary, the forward equations describing the evolution of the ice surface and ice velocities based on an SSA model (in which $\boldsymbol{u}$ is not divergence-free) read

$$h_t + \nabla \cdot (\boldsymbol{u}H) = a, \ t \geq 0, \ \boldsymbol{x} \in \omega,$$

$$h(\boldsymbol{x}, 0) = h_0(\boldsymbol{x}), \ \boldsymbol{x} \in \omega, \quad h(\boldsymbol{x}, t) = h_\gamma(\boldsymbol{x}, t), \ \boldsymbol{x} \in \gamma_u,$$

$$\nabla \cdot \varsigma - Cf(\boldsymbol{u})\boldsymbol{u} = \rho g H \nabla h, \ \boldsymbol{x} \in \omega, \tag{8}$$

$$\boldsymbol{n} \cdot \boldsymbol{u}(\boldsymbol{x}, t) = u_u(\boldsymbol{x}, t), \boldsymbol{x} \in \gamma_u, \quad \boldsymbol{n} \cdot \boldsymbol{u}(\boldsymbol{x}, t) = u_d(\boldsymbol{x}, t), \boldsymbol{x} \in \gamma_d,$$

$$\boldsymbol{t} \cdot \varsigma \boldsymbol{n} = -C_\gamma f_\gamma(\boldsymbol{t} \cdot \boldsymbol{u}) \boldsymbol{t} \cdot \boldsymbol{u}, \ \boldsymbol{x} \in \gamma_g, \quad \boldsymbol{t} \cdot \varsigma \boldsymbol{n} = 0, \ \boldsymbol{x} \in \gamma_w.$$

Above, $\boldsymbol{t}$ is the tangential vector on $\gamma = \gamma_u \cup \gamma_d$ such that $\boldsymbol{n} \cdot \boldsymbol{t} = 0$. The inflow and outflow normal velocities $u_u \leq 0$ and

$u_d > 0$ are specified on $\gamma_u$ and $\gamma_d$. The lateral side of the ice $\gamma$ is split into $\gamma_g$ and $\gamma_w$ with $\gamma = \gamma_g \cup \gamma_w$. There is friction in the tangential direction on $\gamma_g$ which depends on the tangential velocity $\boldsymbol{t} \cdot \boldsymbol{u}$ with the friction coefficient $C_\gamma$ and friction function $f_\gamma$. There is no friction on the wet boundary $\gamma_w$.

For ice sheets that develop an ice shelf, the latter is assumed to be at hydrostatic equilibrium. In such a case, a calving front boundary condition (Schoof, 2007; van der Veen, 1996) is applied at $\gamma_d$, in the form of the depth integrated stress balance ($\rho_w$

is the density of seawater)

$$\varsigma(\boldsymbol{u}) \cdot \boldsymbol{n} = \frac{1}{2} \rho g H^2 \left( 1 - \frac{\rho}{\rho_w} \right) \boldsymbol{n}, \ \boldsymbol{x} \in \gamma_d. \tag{9}$$





With (9), a calving rate $u_c$ can be determined at the ice front, see Sect. 3.2.1.

## 3 Adjoint equations

In this section, the adjoint equations are discussed, as emerging in a FS framework (Sect. 3.1) and in a SSA framework (Sect. 3.2), respectively. The adjoint equations follow from the Lagrangian of the forward equations after partial integration.

Lengthy derivations have been moved to Appendix A.

On the ice surface $\Gamma_s$ and over the time interval $[0, T]$, we consider the functional $\mathcal{F}$

$$\mathcal{F} = \int_0^T \int_{\Gamma_s} F(\boldsymbol{u}, h) \, \mathrm{d}\boldsymbol{x} \, \mathrm{d}t \,. \tag{10}$$

We wish to determine its sensitivity to perturbations in both the friction coefficient $C(\boldsymbol{x}, t)$ at the base of the ice, and the topography $b(\boldsymbol{x})$ of the base itself. We distinguish two cases: either $\boldsymbol{u}$ and $h$ satisfy the FS equations (4), or the SSA equations

(8). Given $\mathcal{F}$, its forward solution $(\boldsymbol{u}, p, h)$ to (4) or $(\boldsymbol{u}, h)$ to (8), and its adjoint solution $(\boldsymbol{v}, q, \psi)$ or $(\boldsymbol{v}, \psi)$ to the adjoint FS and adjoint SSA equations (both derived in the following and in Appendix A), we introduce a Lagrangian $\mathcal{L}(\boldsymbol{u}, p, h; \boldsymbol{v}, q, \psi; b, C)$. The effect of perturbations $\delta C$ and $\delta b$ in $C$ and $b$ on $\mathcal{F}$ is given by the perturbation $\delta \mathcal{L}$, viz.

$$\delta \mathcal{F} = \delta \mathcal{L} = \mathcal{L}(\boldsymbol{u} + \delta\boldsymbol{u}, p + \delta p, h + \delta h; \boldsymbol{v} + \delta\boldsymbol{v}, q + \delta q, \psi + \delta\psi; b + \delta b, C + \delta C) - \mathcal{L}(\boldsymbol{u}, p, h; \boldsymbol{v}, q, \psi; b, C). \tag{11}$$

Examples of $F(\boldsymbol{u}, h)$ in (10) are $F = \|\boldsymbol{u} - \boldsymbol{u}_{\mathrm{obs}}\|^2$, and $F = |h - h_{\mathrm{obs}}|^2$ in an inverse problem, in which the task is to find $b$ and

$C$ such that they match observations $\boldsymbol{u}_{\mathrm{obs}}$ and $h_{\mathrm{obs}}$ at the surface $\Gamma_s$, see also Gillet-Chaulet et al. (2016); Isaac et al. (2015); Morlighem et al. (2013); Petra et al. (2012). Another example is $F(\boldsymbol{u}, h) = \frac{1}{T} u_1(\boldsymbol{x}, t) \delta(\boldsymbol{x} - \boldsymbol{x}_*)$ with the Dirac delta $\delta$ at $\boldsymbol{x}_*$ to measure the time averaged horizontal velocity $u_1$ at $\boldsymbol{x}_*$ on the ice surface $\Gamma_s$ with

$$\mathcal{F} = \int_0^T \int_{\Gamma_s} F(\boldsymbol{u}, h) \, \mathrm{d}\boldsymbol{x} \, \mathrm{d}t = \frac{1}{T} \int_0^T u_1(\boldsymbol{x}_*, t) \, \mathrm{d}t,$$

where $T$ is the duration of the observation at $\Gamma_s$. If the horizontal velocity is observed at $(x_*, t_*)$ then $F(\boldsymbol{u}, h) = u_1(\boldsymbol{x}, t)\delta(\boldsymbol{x} - $

$\boldsymbol{x}_*)\delta(t - t_*)$ and

$$\mathcal{F} = \int_0^T \int_{\Gamma_s} F(\boldsymbol{u}, h) \, \mathrm{d}\boldsymbol{x} \, \mathrm{d}t = u_1(\boldsymbol{x}_*, t_*). \tag{12}$$

The sensitivity in $\mathcal{F}$ and $u_1$ in (12) to perturbations in $C$ and $b$ is then given by (11) with the forward and adjoint solutions. The same forward and adjoint equations are solved both for the inverse problem and the sensitivity problem but with different forcing function $F$. In the inverse problem, the change $\delta \mathcal{F}$ in $\mathcal{F}$ is of interest when $\delta C$ and $\delta b$ are changed during the solution

procedure. In order to minimize $\mathcal{F}$, $\delta C$ and $\delta b$ are chosen such that $\delta \mathcal{F} < 0$ and $\mathcal{F}$ decreases. This procedure is repeated iteratively until $\delta \mathcal{F} \to 0$ and $\mathcal{F}$ has reached a minimum.





### 3.1 Adjoint equations based on the FS model

In this section, we introduce the adjoint equations and the perturbation of the Lagrangian function. The detailed derivations of (13) and (16) below are given in Appendix A, starting from the weak form of the FS equations (4) on $\Omega$, and by using integration by parts, and applying boundary conditions as in Martin and Monnier (2014); Petra et al. (2012).

The definition of the Lagrangian $\mathcal{L}$ for the FS equations is given in (A15) in Appendix A where $(\boldsymbol{v}, q, \psi)$ are the Lagrange multipliers corresponding to the forward equations for $(\boldsymbol{u}, p, h)$. To determine $(\boldsymbol{v}, q, \psi)$, the following adjoint problem is solved:

$$
\begin{aligned}
&\psi_t + \nabla \cdot (\boldsymbol{u}\psi) - \boldsymbol{h} \cdot \boldsymbol{u}_z \psi = F_h + F_{\boldsymbol{u}} \cdot \boldsymbol{u}_z, \text{ on } \Gamma_s, \ 0 \le t \le T, \\
&\psi(\boldsymbol{x}, T) = 0, \ \psi(\boldsymbol{x}, t) = 0, \text{ on } \Gamma_d, \\
&-\nabla \cdot \tilde{\boldsymbol{\sigma}}(v, q) = -\nabla \cdot (2\tilde{\boldsymbol{\eta}}(\boldsymbol{u}) \star \mathbf{D}(\boldsymbol{v})) + \nabla q = \mathbf{0}, \quad \nabla \cdot \boldsymbol{v} = 0, \text{ in } \Omega(t), \\
&\tilde{\boldsymbol{\sigma}}(v, q)\boldsymbol{n} = -(F_{\boldsymbol{u}} + \psi \boldsymbol{h}), \text{ on } \Gamma_s, \\
&\mathbf{T}\tilde{\boldsymbol{\sigma}}(v, q)\boldsymbol{n} = -Cf(\mathbf{T}\boldsymbol{u})(\mathbf{I} + \mathbf{F}_b(\mathbf{T}\boldsymbol{u}))\mathbf{T}\boldsymbol{v}, \text{ on } \Gamma_b, \\
&\boldsymbol{n} \cdot \boldsymbol{v} = 0, \text{ on } \Gamma_b,
\end{aligned}
\tag{13}
$$

where the derivatives of $F$ with respect to $\boldsymbol{u}$ and $h$ are

$$
F_{\boldsymbol{u}} = \left( \frac{\partial F}{\partial u_1}, \frac{\partial F}{\partial u_2}, \frac{\partial F}{\partial u_3} \right)^T, \quad F_h = \frac{\partial F}{\partial h}.
$$

Note that (13) consists of equations for the adjoint elevation $\psi$, the adjoint velocity $\boldsymbol{v}$, and the adjoint pressure $q$. Compared to the steady state adjoint equation for the FS equations discussed in Petra et al. (2012), an advection equation is added in (13) for the Lagrange multiplier $\psi(\boldsymbol{x}, t)$ on $\Gamma_s$ with a right hand side depending on the observation function $F$ and one term depending on $\psi$ in the boundary condition on $\Gamma_s$. The adjoint elevation equation for $\psi$ can be solved independently of the

adjoint stress equation since it is independent of $\boldsymbol{v}$. If $h$ is observed and $F_h \neq 0$ and $F_{\boldsymbol{u}} = \mathbf{0}$, then the adjoint elevation equation must be solved together with the adjoint stress equation. Otherwise, the term $\psi \boldsymbol{h}$ vanishes in the right hand side of the boundary condition of the adjoint stress equation and the solution is $\boldsymbol{v} = \mathbf{0}$ with $\delta \mathcal{F} = 0$ in (16), see below. The adjoint viscosity and adjoint stress are

$$
\begin{aligned}
\tilde{\boldsymbol{\eta}}(\boldsymbol{u}) &= \eta(\boldsymbol{u}) \left( \mathcal{I} + \tfrac{1-n}{n\mathbf{D}(\boldsymbol{u}):\mathbf{D}(\boldsymbol{u})} \mathbf{D}(\boldsymbol{u}) \otimes \mathbf{D}(\boldsymbol{u}) \right), \\
\tilde{\boldsymbol{\sigma}}(v, q) &= 2\tilde{\boldsymbol{\eta}}(\boldsymbol{u}) \star \mathbf{D}(\boldsymbol{v}) - q\mathbf{I},
\end{aligned}
\tag{14}
$$

cf. also Petra et al. (2012). For the rank four-tensor $\mathcal{I}$, $\mathcal{I}_{ijkl} = 1$ only when $i = j = k = l$, otherwise $\mathcal{I}_{ijkl} = 0$. The $\star$ operation in (14) between a rank four-tensor $\mathcal{A}$ and a rank two-tensor (viz., a matrix) $\mathbf{C}$ is defined by $(\mathcal{A} \star \mathbf{C})_{ij} = \sum_{kl} \mathcal{A}_{ijkl} C_{kl}$. In general, $\mathbf{F}_b(\mathbf{T}\boldsymbol{u})$ in (13) is a linearization of the friction law $f(\mathbf{T}\boldsymbol{u})$ in (4) with respect to the variable $\mathbf{T}\boldsymbol{u}$. For instance, with a Weertman type friction law, $f(\mathbf{T}\boldsymbol{u}) = \|\mathbf{T}\boldsymbol{u}\|^{m-1}$,

$$
\mathbf{F}_b(\mathbf{T}\boldsymbol{u}) = \frac{m-1}{\mathbf{T}\boldsymbol{u} \cdot \mathbf{T}\boldsymbol{u}}(\mathbf{T}\boldsymbol{u}) \otimes (\mathbf{T}\boldsymbol{u}).
\tag{15}
$$





The perturbation of the Lagrangian function with respect to a perturbation $\delta C$ in the slip coefficient $C(\boldsymbol{x},t)$ involves the tangential components of the forward and adjoint velocities, $\mathbf{T}\boldsymbol{u}$ and $\mathbf{T}\boldsymbol{v}$ at the ice base $\Gamma_b$, and is given by:

$$\delta\mathcal{F} = \delta\mathcal{L} = \int_0^T \int_{\Gamma_b} f(\mathbf{T}\boldsymbol{u})\mathbf{T}\boldsymbol{u}\cdot\mathbf{T}\boldsymbol{v}\,\delta C\,\mathrm{d}\boldsymbol{x}\,\mathrm{d}t. \tag{16}$$

### 3.1.1 Time-dependent perturbations

Let us now investigate the effect of time-dependent perturbations in the friction parameter on modelled ice velocities and ice surface elevation. A numerical example, based on the Marine Ice Sheet Model Intercomparison Project (MISMIP) (Pattyn et al., 2012) used also in Cheng and Lötstedt (2020) illustrates the findings presented.

Suppose that the velocity component $u_{1*} = u_1(\boldsymbol{x}_*,t_*)$ is observed at $(\boldsymbol{x}_*,t_*)$ at the ice surface as in (12) and that $t_* < T$, then

$$u_{1_*} = \mathcal{F} = \int_0^T \int_{\Gamma_s} F(\boldsymbol{u})\,\mathrm{d}\boldsymbol{x}\,\mathrm{d}t,$$

with $F(\boldsymbol{u}) = u_1\delta(\boldsymbol{x}-\boldsymbol{x}_*)\delta(t-t_*)$, $F_{u_1} = \delta(\boldsymbol{x}-\boldsymbol{x}_*)\delta(t-t_*)$, $F_{u_2} = F_{u_3} = 0$, $F_h = 0$. Above, we have introduced the simplifying notation that a variable with subscript $*$ is a short-hand for it being evaluated at $(\boldsymbol{x}_*,t_*)$, or, if it is time independent, at $\boldsymbol{x}_*$.

The procedure to determine the sensitivity is as follows. First, the forward equation (4) is solved for $\boldsymbol{u}(\boldsymbol{x},t)$ from $t = 0$ to $t = T$. Then, the adjoint equation (13) is solved backward in time (from $t = T$ to $t = 0$) with $\psi(\boldsymbol{x},T) = 0$ as the corresponding

final condition. Obviously, the solution for $t_* < t \leq T$ is $\psi(\boldsymbol{x},t) = 0$ and $\boldsymbol{v}(\boldsymbol{x},t) = \boldsymbol{0}$. Letting $\boldsymbol{e}^i$ denote the unit vector with 1 in the $i$:th component, the boundary condition in (13) becomes

$$\tilde{\boldsymbol{\sigma}}(\boldsymbol{v},q)\boldsymbol{n} = -\boldsymbol{e}^1\delta(\boldsymbol{x}-\boldsymbol{x}_*)\delta(t-t_*) - \psi\boldsymbol{h}$$

at $t = t_*$. For $t < t_*$, $\tilde{\boldsymbol{\sigma}}(\boldsymbol{v},q)\boldsymbol{n} = -\psi\boldsymbol{h}$. Since $\psi$ is small for $t < t_*$ (see Sect. 3.1.3), the dominant part of the solution is $\boldsymbol{v}(\boldsymbol{x},t) = \boldsymbol{v}_0(\boldsymbol{x})\delta(t-t_*)$ for some $\boldsymbol{v}_0$.

We start by investigating the response of ice velocities to perturbations in friction at the base: When the slip coefficient at the ice base is changed by $\delta C$, then the change in $u_{1*}$ at $\Gamma_s$ is, according to (16), given by

$$\delta u_{1*} = \delta\mathcal{L} = \int_0^T \int_{\Gamma_b} f(\mathbf{T}\boldsymbol{u})\mathbf{T}\boldsymbol{u}\cdot\mathbf{T}\boldsymbol{v}\,\delta C\,\mathrm{d}\boldsymbol{x}\,\mathrm{d}t \approx \int_{\Gamma_b} f(\mathbf{T}\boldsymbol{u})\mathbf{T}\boldsymbol{u}\cdot\mathbf{T}\boldsymbol{v}_0\,\delta C(\boldsymbol{x},t_*)\,\mathrm{d}\boldsymbol{x}. \tag{17}$$

This implies that the perturbation $\delta u_{1*}$ mainly depends on $\delta C$ at time $t_*$ and that contributions from previous $\delta C(\boldsymbol{x},t)$, $t < t_*$, are small. If we observe the horizontal velocity, then it responds instantaneously in time to the change in basal friction.

Further, to investigate the response of the ice surface elevation, $h_*$ at $\Gamma_s$, to perturbations in basal friction, one considers

$$F(h) = h(x,t)\delta(\boldsymbol{x}-\boldsymbol{x}_*)\delta(t-t_*),\ F_h = \delta(\boldsymbol{x}-\boldsymbol{x}_*)\delta(t-t_*),\ F_{\boldsymbol{u}} = \boldsymbol{0}.$$

The solution of the adjoint equation (13) with $\tilde{\boldsymbol{\sigma}}(\boldsymbol{v},q)\boldsymbol{n} = -\psi\boldsymbol{h}$ at $\Gamma_s$ for $\boldsymbol{v}(\boldsymbol{x},t)$ is non-zero since $\psi(\boldsymbol{x},t) \neq 0$ for $t < t_*$.



In applied scenarios, friction at the base of an ice sheet is expected to exhibit seasonal variations. These can be expressed by $\delta C(\boldsymbol{x}, t) = \delta C_0(\boldsymbol{x}) \cos(2\pi t/\tau)$, viz. a time dependent perturbation added to a stationary time average $C(\boldsymbol{x})$, with $0 < \delta C_0 \leq C$. If, for illustrational purposes, $\tau = 1$ (so, one year, from January to December), then Northern hemisphere cold and warm seasons can in a simplified manner be associated with $n\tau$, $n = 0, 1, 2, \dots$ (winter) and $(n + 1/2)\tau$, $n = 0, 1, 2, \dots$ (summer).

Assume further that $f(\mathbf{T}\boldsymbol{u})\mathbf{T}\boldsymbol{u} \cdot \mathbf{T}\boldsymbol{v}$ is approximately constant in time. This is the case if $\boldsymbol{u}$ varies slowly. Then $\psi \approx \text{const}$ and $\boldsymbol{v} \approx \text{const}$ for $t < t_*$. The change in ice surface elevation, $\delta h$, due to time-dependent variations in basal friction varies as

$$
\begin{aligned}
\delta h_* & = \delta \mathcal{L} = \int\limits_0^T \int\limits_{\Gamma_b} f(\mathbf{T}\boldsymbol{u})\mathbf{T}\boldsymbol{u} \cdot \mathbf{T}\boldsymbol{v} \, \delta C(\boldsymbol{x}, t) \, \mathrm{d}\boldsymbol{x} \, \mathrm{d}t \\
& \approx \int\limits_0^{t_*} \cos(2\pi t/\tau) \, \mathrm{d}t \int\limits_{\Gamma_b} f(\mathbf{T}\boldsymbol{u})\mathbf{T}\boldsymbol{u} \cdot \mathbf{T}\boldsymbol{v} \, \delta C_0 \, \mathrm{d}\boldsymbol{x} = \frac{\tau}{2\pi} \sin(2\pi t_*/\tau) \int\limits_{\Gamma_b} f(\mathbf{T}\boldsymbol{u})\mathbf{T}\boldsymbol{u} \cdot \mathbf{T}\boldsymbol{v} \, \delta C_0 \, \mathrm{d}\boldsymbol{x}.
\end{aligned}
\tag{18}
$$

Obviously, from the properties of the cosine function, the friction perturbation $\delta C$ is large at $t_* = 0, \tau/2, \tau \dots$, and vanishes at $t_* = \tau/4, 3\tau/4, \dots$. Yet, (18) shows that $\delta h_* = 0$ at $t_* = 0, \tau \dots$ (so, during maximal friction in the winter) and at $t_* =$

$\tau/2, 3\tau/2 \dots$ (so, during minimal friction in the summer), while $\delta h_* \neq 0$ when $\delta C = 0$ at $t_* = \tau/4, 3\tau/4, \dots$ in the spring and the fall. The response in $h$ by changing $C$ is delayed in phase by $\pi/2$ or in time by $\tau/4 = 0.25$ yr. This is in contrast to the observation of $u_1$ in (17) where a perturbation in $C$ is directly visible.

Particularly in an inverse problem where the phase shift between $\delta C$ and $\delta h$ in (18) is not accounted for, if $h_*$ is measured in the summer with $\delta h(\boldsymbol{x}, t_*) = 0$, then the wrong conclusion would be drawn that there is no change in $C$.

To illustrate this, a two-dimensional numerical example is shown in Fig. 2, where the time scale and friction coefficient are chosen as follows: $\tau = 1$ yr, $\delta C(x, t) = 0.01 C \cos(2\pi t)$ in $x \in [0.9, 1.0] \times 10^6$ m. The ice sheet flows from $x = 0$ to $L = 1.6 \times 10^6$ m on a single slope bed defined by $b(x) = 720 - \frac{778.5x}{7.5 \times 10^5}$. $u_1$ and $h$ are observed at $x \in [0.85, 1.02] \times 10^6$ m. The steady state solution with the GL located at $x_{GL} = 1.035 \times 10^6$ m is perturbed by $\delta \boldsymbol{u}$ and $\delta h$ when $C$ is perturbed by $\delta C$. After perturbation, the GL position will oscillate. For additional details of the setup, we refer to the MISMIP (Pattyn et al., 2012) test

case EXP 1 (step 1) used already in Cheng and Lötstedt (2020). The ice sheet is simulated by FS with Elmer/Ice (Gagliardini et al., 2013) for 10 years.

Fig. 2 shows that the perturbations $\delta u_1$ and $\delta h$ in the grounded part of the ice sheet, specifically at $x_* = 0.85, 0.9, 0.95, 1.0$, and $1.02 \times 10^6$ m for which individual panels are shown, oscillate regularly with a period of 1 year. The perturbations are small outside the interval $[0.9, 1.0] \times 10^6$. The initial condition at $t = 0$ is the steady state solution of the MISMIP problem and the

FS solution with a variable $C$ is essentially that steady state solution plus a small oscillatory perturbation, as in Fig. 2.

The weight $f(\mathbf{T}\boldsymbol{u})\mathbf{T}\boldsymbol{u} \cdot \mathbf{T}\boldsymbol{v}_0$ in (17) is negative and an increase in the friction, $\delta C > 0$, leads to a decrease in the velocity and $\delta C < 0$ increases the velocity in all panels of Fig. 2. The velocities $\delta u_1$ and the surface elevations $\delta h$ are separated by a phase shift in time, $\Delta \phi = \pi/2$, as predicted by (17) and (18).

The weight in (18) for $\delta C_0$ in the integral over $\boldsymbol{x}$ changes sign when the observation point is passing from $x_* = 0.9 \times 10^6$ to

$1.0 \times 10^6$ explaining why the shift changes sign in the red dashed lines shown in the two lower panels of Fig. 2.



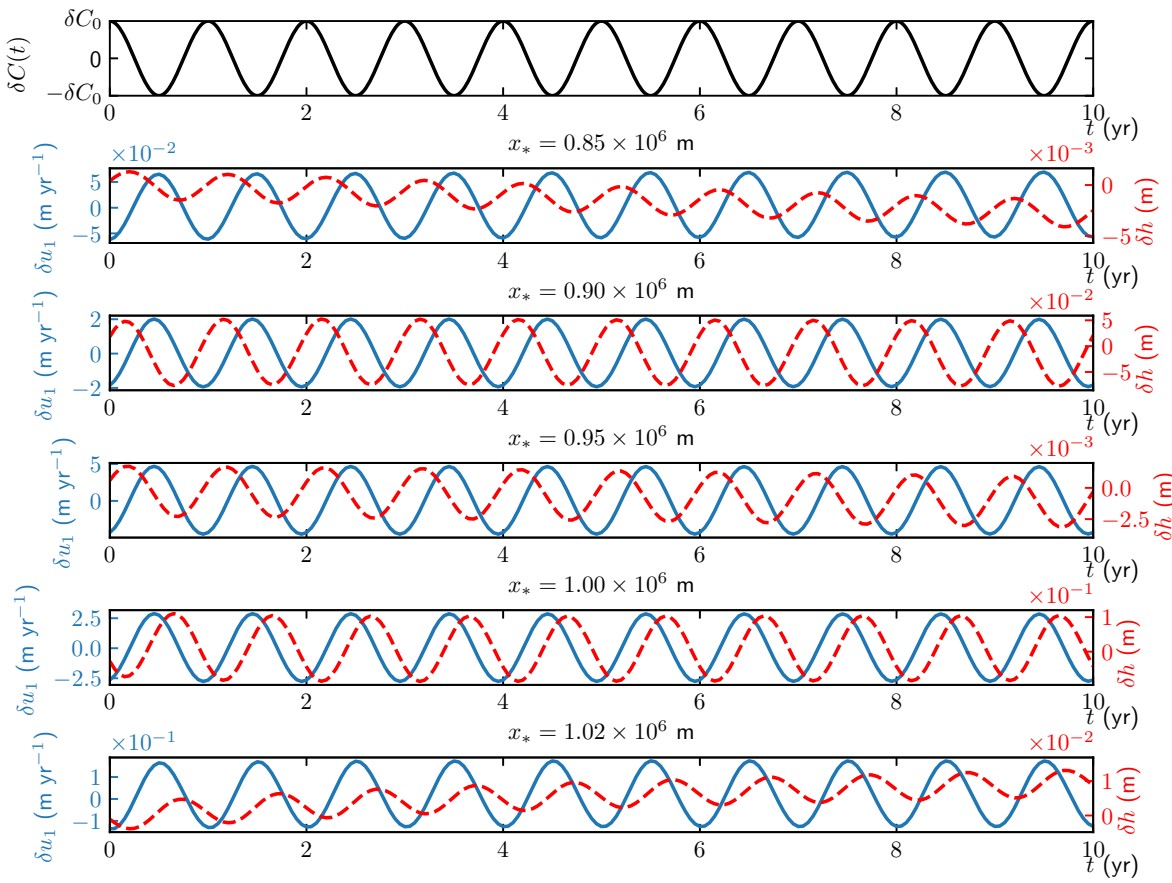

**Figure 2.** Observations at $x_* = 0.85, 0.9, 0.95, 1.0, 1.02 \times 10^6$ m with FS in time $t \in [0, 10]$ of $\delta u_1$ (solid blue) and $\delta h$ (dashed red) with perturbation $\delta C(t) = 0.01 C \cos(2\pi t)$ for $x \in [0.9, 1.0] \times 10^6$ m. Notice the different scales on the $y$-axes.

### 3.1.2 The sensitivity problem and the inverse problem.

From a theoretical point of view, it is interesting to note that there is a relation between the sensitivity problem where the effect of perturbed parameters in the forward model is estimated and the inverse problem used to infer 'unobservable' parameters such as basal friction from observable data, e.g. ice velocity at the ice sheet surface. The same adjoint equations (13) are solved

5 in both problems but with different driving functions defined by $F(\boldsymbol{u}, h)$ in (10).

Let $(\boldsymbol{v}^i, q^i, \psi^i), i = 1, \ldots, d$, be the solution to (13) when $u_i$ is observed at $\bar{\boldsymbol{x}}$ and $F_{\boldsymbol{u}} = \boldsymbol{e}^i \delta(\boldsymbol{x} - \bar{\boldsymbol{x}})$. The adjoint solution and the variation $\delta \mathcal{F}$ of the inverse problem can be expressed in $(\boldsymbol{v}^i, q^i, \psi^i)$.

It is shown in Appendix B that

$$\left( \int_\omega \sum_{i=1}^d w_i(\bar{\boldsymbol{x}}) \boldsymbol{v}^i \, \mathrm{d}\bar{\boldsymbol{x}}, \int_\omega \sum_{i=1}^d w_i(\bar{\boldsymbol{x}}) q^i \, \mathrm{d}\bar{\boldsymbol{x}}, \int_\omega \sum_{i=1}^d w_i(\bar{\boldsymbol{x}}) \psi^i \, \mathrm{d}\bar{\boldsymbol{x}} \right) \tag{19}$$





is a solution of (13) with arbitrary weights $w_i(\bar{\boldsymbol{x}})$, $i = 1, \ldots, d$, when

$$F_{\boldsymbol{u}} = \int_\omega \sum_{i=1}^d w_i(\bar{\boldsymbol{x}}) e^i \delta(\boldsymbol{x} - \bar{\boldsymbol{x}}) \, \mathrm{d}\bar{\boldsymbol{x}} = \sum_{i=1}^d w_i(\bar{\boldsymbol{x}}) e^i. \tag{20}$$

When $C$ is perturbed, the first variation of the functional in (16) is

$$\delta\mathcal{F} = \int_{\Gamma_b} f(\mathbf{T}\boldsymbol{u}) \mathbf{T}\boldsymbol{u} \cdot \mathbf{T}\left( \int_\omega \sum_{i=1}^d w_i(\bar{\boldsymbol{x}}) \boldsymbol{v}^i \, \mathrm{d}\bar{\boldsymbol{x}} \right) \delta C \, \mathrm{d}\boldsymbol{x}. \tag{21}$$

In the inverse problem in Petra et al. (2012),

$$\mathcal{F} = \frac{1}{2} \int_\omega \|\boldsymbol{u}(\boldsymbol{x}) - \boldsymbol{u}_{\mathrm{obs}}(\boldsymbol{x})\|^2 \, \mathrm{d}\boldsymbol{x}, \quad F_{\boldsymbol{u}} = \boldsymbol{u}(\boldsymbol{x}) - \boldsymbol{u}_{\mathrm{obs}}(\boldsymbol{x}). \tag{22}$$

The weights in (20) are $w_i(\boldsymbol{x}) = u_i(\boldsymbol{x}) - u_{\mathrm{obs},i}(\boldsymbol{x})$ and the effect of $\delta C$ on $\mathcal{F}$ is by (21)

$$\delta\mathcal{F} = \int_{\Gamma_b} f(\mathbf{T}\boldsymbol{u}) \mathbf{T}\boldsymbol{u} \cdot \mathbf{T}\tilde{\boldsymbol{v}}(\boldsymbol{x}) \, \delta C \, \mathrm{d}\boldsymbol{x}, \tag{23}$$

where $\tilde{\boldsymbol{v}}$ is a weighted sum of the solutions of the sensitivity problem $\boldsymbol{v}^i$ over the whole domain $\omega$

$$\tilde{\boldsymbol{v}}(\boldsymbol{x}) = \int_\omega \sum_{i=1}^d (u_i(\bar{\boldsymbol{x}}) - u_{\mathrm{obs},i}(\bar{\boldsymbol{x}})) \boldsymbol{v}^i \, \mathrm{d}\bar{\boldsymbol{x}}. \tag{}$$

The same construction of the solution is possible when $h_{\mathrm{obs}}$ is given. Then $d = 1$, $F(h) = \frac{1}{2}(h - h_{\mathrm{obs}})^2$, and $F_h = w = h - h_{\mathrm{obs}}$.
If we are interested in solving the inverse problem and determine $\delta\mathcal{F}$ in (21) in order to iteratively compute the optimal solution
with a gradient method, then we solve (13) directly with $F_{\boldsymbol{u}} = \boldsymbol{u} - \boldsymbol{u}_{\mathrm{obs}}$ or $F_h = h - h_{\mathrm{obs}}$ to obtain $\tilde{\boldsymbol{v}}$ without computing $\boldsymbol{v}^i$.

### 3.1.3   Steady state solution to the adjoint elevation equation in two dimensions.

A further theoretical consideration shows that the solution $\psi$ to the adjoint elevation equation need not be computed to estimate
perturbations in the velocity for a two-dimensional vertical ice sheet at steady state. We show with the analytical solution in
the FS model that the influence of $\psi$ is negligible. It is sufficient to solve the adjoint stress equation for $\boldsymbol{v}$ to estimate the
perturbation in the velocity.

The adjoint steady state equation in a two dimensional vertical ice in (13) is

$$(u_1 \psi)_x = F_h + (\boldsymbol{h}\psi + F_{\boldsymbol{u}}) \cdot \boldsymbol{u}_z, \; z = h, \; 0 \le x \le L. \tag{24}$$

The velocity from the forward equation is $\boldsymbol{u}(x, z) = (u_1, u_3)^T$ and the adjoint elevation $\psi$ satisfies the right boundary condition
$\psi(L) = 0$.





The analytical solution $\psi$ to (24) is derived in Appendix C. Let $g(x) = u_{1z}(x)$ if $u_1$ is observed and let $g(x) = 1$ if $h$ is observed. Then the adjoint solution is

$$\psi(x) = \begin{cases} -\dfrac{g(x_*)}{u_1(x)} \exp\left( -\int\limits_x^{x_*} \dfrac{\boldsymbol{h} \cdot \boldsymbol{u}_z(y)}{u_1(y)}\, \mathrm{d}y \right), & 0 \leq x \leq x_*, \\[4mm] 0, & x_* < x \leq L. \end{cases} \tag{25}$$

So, this solution has a jump $-g(x_*)/u_1(x_*)$ at $x_*$.

With a small $\boldsymbol{h} \cdot \boldsymbol{u}_z(y) \approx 0$ in (25), an approximate solution is $\psi(x) \approx -g(x_*)/u_1(x)$. If $u_1$ is observed and $g(\boldsymbol{x}) = u_{1z} \approx 0$, then $\psi(\boldsymbol{x}) \approx 0$ in (25) and $\psi\boldsymbol{h} \approx 0$ in (13). This is the case in the SSA of the FS model where $u_{1z}(x) = 0$ and in the SIA of the FS equations where $u_{1z}(x,h) = 0$ (Greve and Blatter, 2009; Hutter, 1983). When these approximations are accurate then $u_{1z}$ will be small. Consequently, when $u_1$ is observed, the effect on $\boldsymbol{v}$ in the adjoint stress equation of the solution $\psi$ of the adjoint advection equation in (13) is small. Solving only the adjoint stress equation for $\boldsymbol{v}$ as in Gillet-Chaulet et al. (2016); Isaac et al.

(2015); Petra et al. (2012) yields an adequate answer. The situation is different when $h$ is observed and $\psi \neq 0$ in (25).

## 3.2    Shallow shelf approximation

Starting from (8), a Lagrangian $\mathcal{L}$ of the SSA equations is defined, using the technique described and applied to the FS equations in Petra et al. (2012). By evaluating $\mathcal{L}$ at the forward solution $(\boldsymbol{u}, h)$ and at the adjoint solution $(\boldsymbol{v}, \psi)$, the adjoint SSA equations are obtained. Then, the effect of perturbed data at the ice base manifests itself at the ice surface as a perturbation $\delta\mathcal{L}$; for details,

see Appendix A. The adjoint SSA equations read:

$$\begin{aligned} &\psi_t + \boldsymbol{u} \cdot \nabla\psi + 2\eta\mathbf{B}(\boldsymbol{u}) : \mathbf{D}(\boldsymbol{v}) - \rho g H \nabla \cdot \boldsymbol{v} + \rho g \boldsymbol{v} \cdot \nabla b = F_h, \ \text{in } \omega, \ 0 \leq t \leq T, \\ &\psi(\boldsymbol{x}, T) = 0, \ \text{in } \omega, \quad \psi(\boldsymbol{x}, t) = 0, \ \text{on } \gamma_w, \\ &\nabla \cdot \tilde{\boldsymbol{\varsigma}}(\boldsymbol{v}) - Cf(\boldsymbol{u})(\mathbf{I} + \mathbf{F}_\omega(\boldsymbol{u}))\boldsymbol{v} - H\nabla\psi = -F_{\boldsymbol{u}}, \quad \text{in } \omega, \\ &\boldsymbol{t} \cdot \tilde{\boldsymbol{\varsigma}}(\boldsymbol{v})\boldsymbol{n} = -C_\gamma f_\gamma(\boldsymbol{t} \cdot \boldsymbol{u})(1 + F_\gamma(\boldsymbol{t} \cdot \boldsymbol{u}))\boldsymbol{t} \cdot \boldsymbol{v}, \ \text{on } \gamma_g, \quad \boldsymbol{t} \cdot \tilde{\boldsymbol{\varsigma}}(\boldsymbol{v})\boldsymbol{n} = 0, \ \text{on } \gamma_w, \\ &\boldsymbol{n} \cdot \boldsymbol{v} = 0, \ \text{on } \gamma, \end{aligned} \tag{26}$$

where the adjoint viscosity $\tilde{\eta}$ and adjoint stress $\tilde{\boldsymbol{\varsigma}}$ are (cf. (14) for the case of FS)

$$\begin{aligned} \tilde{\boldsymbol{\eta}}(\boldsymbol{u}) &= \eta(\boldsymbol{u})\left( \mathcal{I} + \tfrac{1-n}{n\mathbf{B}(\boldsymbol{u}):\mathbf{D}(\boldsymbol{u})}\mathbf{B}(\boldsymbol{u}) \otimes \mathbf{D}(\boldsymbol{u}) \right), \\ \tilde{\boldsymbol{\varsigma}}(\boldsymbol{v}) &= 2H\tilde{\boldsymbol{\eta}}(\boldsymbol{u}) \star \mathbf{B}(\boldsymbol{v}). \end{aligned} \tag{27}$$

From (26) it is seen that the adjoint SSA equations have the same structure as the adjoint FS equations (13). There is one stress

equation for the adjoint velocity $\boldsymbol{v}$, and one equation for the Lagrange multiplier $\psi$ corresponding to the surface elevation equation in (8). However, the advection equation for $\psi$ in (26) depends on $\boldsymbol{v}$, implying a fully coupled system for $\boldsymbol{v}$ and $\psi$. Equations (26) are solved backward in time with a final condition on $\psi$ at $t = T$. As in (8), there is no time derivative in the stress equation. With a Weertman friction law, viz. $f(\boldsymbol{u}) = \|\boldsymbol{u}\|^{m-1}$ and $f_\gamma(\boldsymbol{t} \cdot \boldsymbol{u}) = |\boldsymbol{t} \cdot \boldsymbol{u}|^{m-1}$ (cf. also Appendix A1),

$$\mathbf{F}_\omega(\boldsymbol{u}) = \frac{m-1}{\boldsymbol{u} \cdot \boldsymbol{u}} \boldsymbol{u} \otimes \boldsymbol{u}, \quad F_\gamma = m - 1.$$





If the friction coefficient $C$ at the ice base (both where it is grounded on bedrock ($C > 0$) and floating ($C = 0$)) is changed by $\delta C$, if the bottom topography is changed by $\delta b$, and if the lateral friction coefficient $C_\gamma$ is changed by $\delta C_\gamma$, then it follows from Appendix A2 that the Lagrangian $\mathcal{L}$ is changed by (note that the weight in front of $\delta C$ in (28) is actually the same as in (16))

$$\delta\mathcal{L} = \int\limits_0^T \int\limits_\omega (2\eta\mathbf{B}(\boldsymbol{u}) : \mathbf{D}(\boldsymbol{v}) + \rho g\boldsymbol{v}\cdot\nabla h + \nabla\psi\cdot\boldsymbol{u})\,\delta b - f(\boldsymbol{u})\boldsymbol{u}\cdot\boldsymbol{v}\,\delta C\,\mathrm{d}\boldsymbol{x}\,\mathrm{d}t - \int\limits_0^T \int\limits_{\gamma_g} f_\gamma(\boldsymbol{t}\cdot\boldsymbol{u})\boldsymbol{t}\cdot\boldsymbol{u}\,\boldsymbol{t}\cdot\boldsymbol{v}\,\delta C_\gamma\,\mathrm{d}s\,\mathrm{d}t. \tag{28}$$

The same perturbations in $\delta C, \delta b$, and $\delta C_\gamma$ could be allowed for the FS equations in (16) but because the FS equations are more complicated than the SSA equations, the complexity of the derivation in the appendix and the expression for $\delta\mathcal{L}$ would increase considerably, which is why we refrain from considering them here.

Suppose that only $h$ is observed with $F_h \neq 0$ and $F_{\boldsymbol{u}} = 0$ in (26). Then the adjoint elevation equation must be solved for $\psi \neq 0$ to have a $\boldsymbol{v} \neq \boldsymbol{0}$ in the adjoint stress equation and a perturbation in the Lagrangian in (28). The same result follows from

the adjoint FS equations. If $F_h \neq 0$ and $F_{\boldsymbol{u}} = 0$ in (13), then $\psi \neq 0$. Consequently, $\boldsymbol{v} \neq \boldsymbol{0}$ and a perturbation $\delta C$ will cause a perturbation $\delta\mathcal{L}$ in (16). The conclusion that the adjoint elevation equation must be solved if the surface elevation is observed is independent of the two ice models.

In a broader context, it is worth emphasizing that the adjoint equation derived in MacAyeal (1993) is identical to the stress equation in (26), if $H$ is constant, $\mathbf{F}_\omega = 0$ and $\tilde{\boldsymbol{\eta}}(\boldsymbol{u}) = \eta(\boldsymbol{u})$.

### 3.2.1   SSA in two dimensions.

In this section, the forward and adjoint SSA equations are presented for the case of an idealized, two-dimensional vertical sheet in the $x$-$z$ plane, see Fig. 1. In Sect. 3.2.2, numerical examples of the steady state case are given. The forward and adjoint SSA equations are derived from (8) and (26) by letting $H$ and $u_1$ be independent of $y$, and setting $u_2 = 0$. Since there is no lateral force, $C_\gamma = 0$. The position of the grounding line is denoted by $x_{GL}$, and $\Gamma_b = [0, x_{GL}]$, $\Gamma_w = (x_{GL}, L]$. Basal friction $C$ is

positive and constant where the ice sheet is grounded on bedrock, while $C = 0$ at the floating ice shelves' lower boundary. To simplify notation, we let $u = u_1$ and $v = v_1$. The forward equations thus become

$$\begin{aligned}
&h_t + (uH)_x = a, \ 0 \le t \le T, 0 \le x \le L, \\
&h(x,0) = h_0(x), \ h(0,t) = h_L(t), \\
&(H\eta u_x)_x - Cf(u)u - \rho gHh_x = 0, \\
&u(0,t) = u_u(t), \ u(L,t) = u_d(t),
\end{aligned} \tag{29}$$

where $u_u$ is the speed of the ice flux at $x = 0$ and $u_d$ is the calving speed at $x = L$. If $x = 0$ is at the ice divide, then $u_u = 0$. By the stress balance (9), the calving front satisfies

$u_x(L,t) = A\left[\dfrac{\rho gH(L,t)}{4}\left(1 - \dfrac{\rho}{\rho_w}\right)\right]^n.$





Assuming that $u > 0$ and $u_x > 0$, the viscosity becomes $\eta = 2A^{-\frac{1}{n}}u_x^{\frac{1-n}{n}}$, and the friction term with a Weertman law turns into $Cf(u)u = Cu^m$. The adjoint equations for $v$ and $\psi$ follow either from simplifying (26), or from (29) and read as follows:

$$
\begin{aligned}
&\psi_t + u\psi_x + (\eta u_x - \rho g H)v_x + \rho g b_x v = F_h, \quad 0 \le t \le T,\, 0 \le x \le L, \\
&\psi(x, T) = 0,\; \psi(L, t) = 0, \\
&(\tfrac{1}{n}\eta H v_x)_x - Cmf(u)v - H\psi_x = -F_u, \\
&v(0, t) = 0,\; v(L, t) = 0.
\end{aligned}
\tag{30}
$$

Note that the viscosity above is multiplied by a factor $1/n$, $n > 0$ which represents an extension of the adjoint SSA in MacAyeal

(1993) where $n = 1$. The effect on the Lagrangian of perturbations $\delta b$ and $\delta C$ is obtained from (28)

$$
\delta\mathcal{L} = \int_0^T \int_0^L (\psi_x u + v_x \eta u_x + v\rho g h_x)\,\delta b - vf(u)u\,\delta C\,\mathrm{d}x\,\mathrm{d}t.
\tag{31}
$$

The weights or sensitivity functions $w_b$ and $w_C$ multiplying $\delta b$ and $\delta C$ in the integral are defined by

$$
w_b(x, t) = \psi_x u + v_x \eta u_x + v\rho g h_x, \quad w_C(x, t) = -vf(u)u.
\tag{32}
$$

### 3.2.2  The two-dimensional forward steady state solution.

We turn now to a discussion of steady-state solutions to the system (30). Except from letting all time derivatives vanish, even the longitudinal stress can be ignored in the steady state solution, see Schoof (2007). Moreover, the viscosity terms in (30) are often small and can hence be neglected, too. The advantage resulting from these simplifications is that both the forward and adjoint equations can be solved analytically on a reduced computational domain where $x \in [0, x_{GL}]$. The analytical approximations are less accurate close to the ice divide where some of the above assumptions are not valid. With a sliding law in the form

$f(u) = u^{m-1}$, (29) thus reduces to

$$
\begin{aligned}
&(uH)_x = a,\; 0 \le x \le x_{GL}, \\
&H(0) = H_0, \\
&-Cu^m - \rho g H h_x = 0, \\
&u(0) = 0,
\end{aligned}
\tag{33}
$$

and the adjoint equations (30) reduce, under the assumption that the basal topography is characterized by a small spatial gradient $b_x$, to

$$
\begin{aligned}
&u\psi_x - \rho g H v_x = F_h,\; 0 \le x \le x_{GL}, \\
&\psi_x(0) = 0,\; \psi(x_{GL}) = 0, \\
&-Cmu^{m-1}v - H\psi_x = -F_u, \\
&v(0) = 0.
\end{aligned}
\tag{34}
$$




The solution to the forward equation (33) is presented for the case when $a$ and $C$ are constant, (for details, see (D3) and (D4) in Appendix D):

$$
\begin{aligned}
H(x) &= \left( H_{GL}^{m+2} + \frac{m+2}{m+1} \frac{Ca^m}{\rho g} \left( x_{GL}^{m+1} - x^{m+1} \right) \right)^{\frac{1}{m+2}}, \, 0 \leq x \leq x_{GL}, \\
H(x) &= H_{GL}, \, x_{GL} < x < L, \\
u(x) &= \frac{ax}{H}, \, 0 \leq x \leq x_{GL}, \quad u(x) = \frac{ax}{H_{GL}}, \, x_{GL} < x < L.
\end{aligned}
\tag{35}
$$

The solution is calibrated with the ice thickness $H_{GL} = H(x_{GL})$ at the fixed grounding line.

Fig. 3 displays solutions from (35) obtained with data from the MISMIP (Pattyn et al., 2012) test case EXP 1 (step 1) chosen in Cheng and Lötstedt (2020). The ice sheet rests on a downward sloping bedrock with constant slope angle and lifts from it at the grounding line position $x_{GL}$. As $x$ approaches $x_{GL}$, $H$ decreases to approach to $H_{GL}$ in Fig. 3(b). The larger the friction coefficient $C$ and accumulation rate $a$ are, the steeper the decrease in $H$ is in (35). The numerator in $u$ increases and the denominator decreases when $x \to x_{GL}$ resulting in a rapid increase in $u$. The MISMIP example is such that the SSA solution

is close to the FS solution. Numerical experiments in Cheng and Lötstedt (2020) show that an accurate solution compared to the FS and SSA solutions is obtained with $u$ and $H$ in (35) solving (33).

Finally, it is noted that an alternative solution to (29) valid for the floating ice shelf, $x > x_{GL}$, but under the restraining assumption of $H(x)$ being linear in $x$, is found in Greve and Blatter (2009).

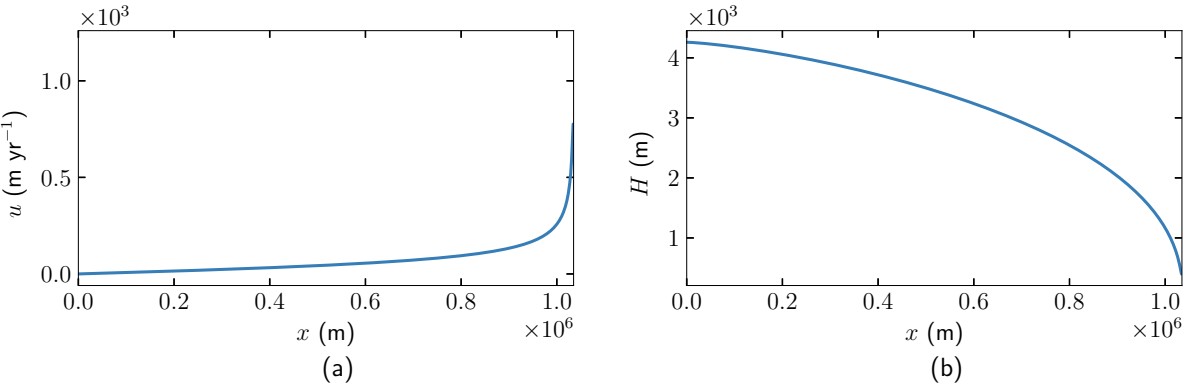

**Figure 3.** The analytical solutions $u(x)$ and $H(x)$ in (35) for a grounded ice in $[0, x_{GL}]$.

### 3.2.3   The two-dimensional adjoint steady state solution with $F_u \neq 0$.

In this section, the analytical solution to the adjoint equation (34) is discussed. The derivation of the solution is detailed in Appendix E to Appendix F. It is here sufficient to recall that the below given solution is derived under the assumptions that $b_x \ll H_x$, and that $a$ and $C$ are constants.



For observations of $u$ at $x_*$,

$$\mathcal{F} = \int_0^L u(x)\delta(x - x_*)\,\mathrm{d}x = u_*, \; F_u = \delta(x - x_*), \; F_h = 0,$$

the adjoint solutions are

$$
\begin{aligned}
\psi(x) &= \frac{Ca^m x_*}{\rho g H_*^{m+3}}(x_{GL}^m - x^m), \; x_* < x \le x_{GL},\\
\psi(x) &= -\frac{1}{H_*} + \frac{Ca^m x_*}{\rho g H_*^{m+3}}(x_{GL}^m - x_*^m), \; 0 \le x < x_*,\\
v(x) &= \frac{ax_*}{\rho g H_*^{m+3}}H^m, \; x_* < x \le x_{GL},\\
v(x) &= 0, \; 0 \le x < x_*,
\end{aligned}
\tag{36}
$$

5  where $\psi(x)$ and $v(x)$ have discontinuities at the observation point $x_*$. The analytical adjoint solutions $\psi(x)$ and $v(x)$ of the MISMIP case as in Fig. 3 at different $x_*$ positions are shown in Fig. 4(a) and Fig. 5(a). In all figures, $m = 1$ in the friction model.

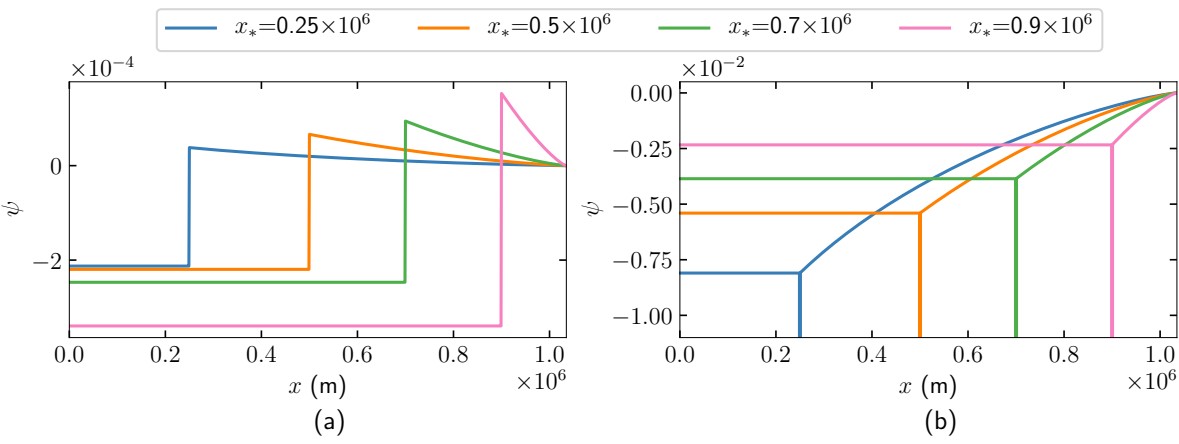

**Figure 4.** The analytical solutions of $\psi$ in (34) of the observations of (a) $u$ and (b) $h$ at different locations $x_* = 0.25 \times 10^6, 0.5 \times 10^6, 0.7 \times 10^6$ and $0.9 \times 10^6$ m.

The perturbation of the Lagrangian (31) is with the Heaviside step function $\mathcal{H}(x)$ and the Dirac delta $\delta(x)$ (cf. Appendix F)

$$
\begin{aligned}
\delta u_* = \delta \mathcal{L} &= \int_0^{x_{GL}} (\psi_x u + v_x \eta u_x + v\rho g h_x)\delta b - vu^m\,\delta C\,\mathrm{d}x\\
&= \int_{x_*^-}^{x_{GL}} \frac{ax_* H^m}{H_*^{m+3}}\left[(m+1)H_x\mathcal{H}(x - x_*) + H\delta(x - x_*)\right]\delta b - \frac{ax_*(ax)^m}{\rho g H_*^{m+3}}\delta C\,\mathrm{d}x\\
&= \frac{u_*\delta b_*}{H_*} - \frac{u_*}{\rho g H_*^{m+2}}\int_{x_*}^{x_{GL}} C(ax)^m\left((m+1)\frac{\delta b}{H} + \frac{\delta C}{C}\right)\,\mathrm{d}x,
\end{aligned}
\tag{37}
$$


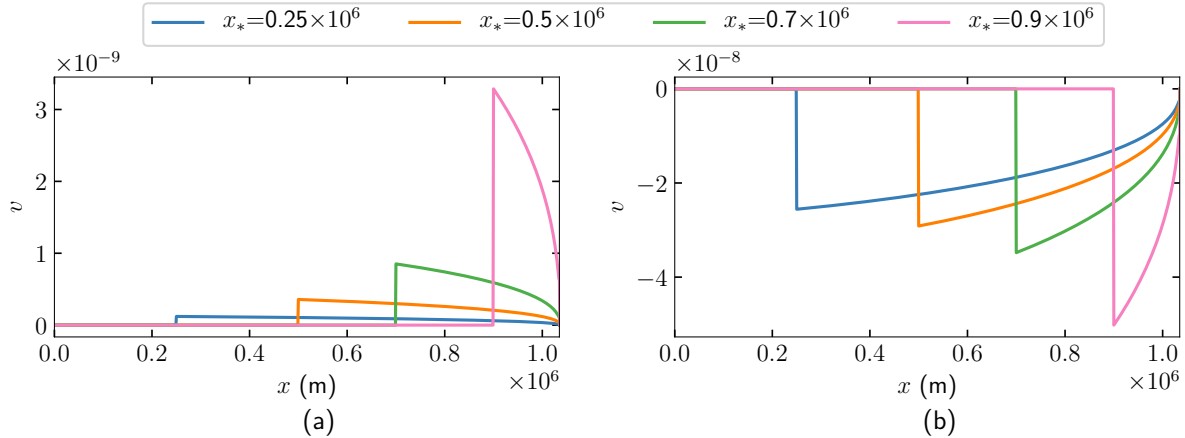

**Figure 5.** The analytical solutions of $v$ in (34) of the observations of (a) $u$ and (b) $h$ at different locations $x_* = 0.25 \times 10^6, 0.5 \times 10^6, 0.7 \times 10^6$ and $0.9 \times 10^6$ m.

or, after scaling with $u_*$:

$$\frac{\delta u_*}{u_*} = \frac{\delta b_*}{H_*} - \frac{1}{\rho g H_*^{m+2}} \int\limits_{x_*}^{x_{GL}} C(ax)^m \left( (m+1)\frac{\delta b}{H} + \frac{\delta C}{C} \right) \, \mathrm{d}x. \tag{38}$$

The weights $w_b$ and $w_C$ in (37) multiplying $\delta b$ and $\delta C$, defined in the same manner as in (31) and (32), are shown in Fig. 6(a) and Fig. 7(a) with the solutions in Fig. 4(a) and Fig. 5(a). The Dirac term is plotted as a vertical line at $x_*$ in Fig. 7(a).

All perturbations in $C$ between $x_*$ and $x_{GL}$ will result in a perturbation of the opposite sign in $u_*$ at the surface because $w_C < 0$ in $(x_*, x_{GL})$ in Fig. 6(a) and (37). The same conclusion holds true for perturbations in $b$ because $w_b < 0$ in $(x_*, x_{GL})$ in Fig. 7(a) but an additional contribution is added from $\delta b$ at $x_*$ by the Dirac delta in $w_b$. A perturbation is less visible in $u$ the farther away from $x_{GL}$ the observation point is since the amplitude of both $w_C$ and $w_b$ decays when $x_*$ decreases.

The relation in (38) between the relative perturbations $\delta b/H, \delta C/C$ and $\delta u/u$ can also be interpreted as a way to quantify the

uncertainty in $u$. An uncertainty $\delta C$ in $C$ and an uncertainty $\delta b$ in $b$ is propagated to an uncertainty $\delta u_*$ in $u$ at $x_*$ by (38). The following conclusions can be drawn from (37) and (38) and Figs. 6 and 7:

(i). The closer perturbations in basal friction are located to the grounding line, the larger perturbations of velocity will be observed at the surface. This is because the weight in front of $\delta C$ increases when $x_* \to x_{GL}$, see Fig. 6, which in turn is an effect of the increasing velocity $u_*$ and the decreasing thickness $H_*$, as the grounding line is approached, see Fig. 3.

Or, compactly expressed, $\delta C$ with support in $[x_*, x_{GL}]$ will cause larger perturbations at the surface the closer $x_*$ is to $x_{GL}$, and the closer $\delta C(x)$ is to $x_{GL}$.

(ii). Variations in the observed velocity $\delta u_*$ at the surface at observation point $x_*$ will include contributions from changes in the frictional parameter, $\delta C$, between $x_*$ and the grounding line $x_{GL}$, and from changes in basal topography, $\delta b$, but it is impossible to disentangle their individual contributions to $\delta u_*$.



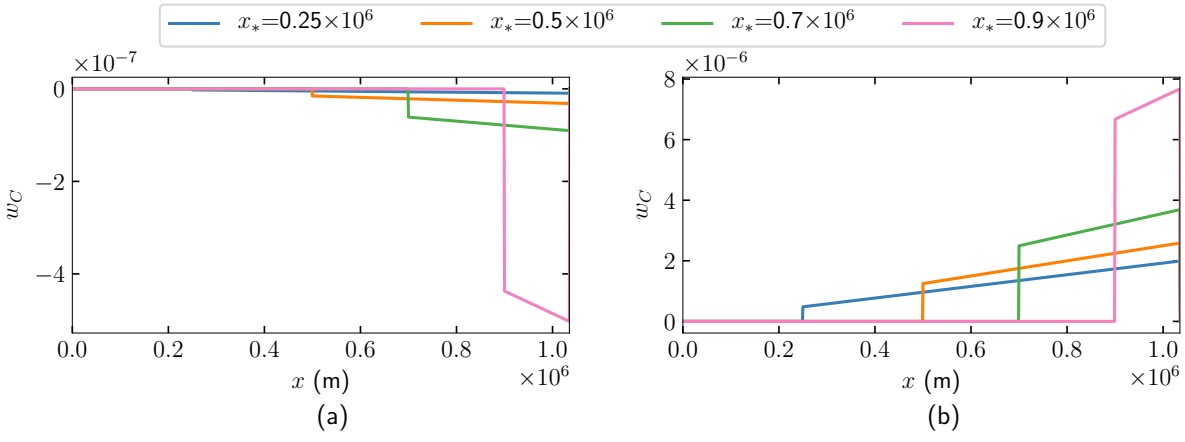

**Figure 6.** The analytical solution of the weights $w_C = -vu^m$ on $\delta C$ in (31) for (a) $u$ and (b) $h$ observed at $x_* = 0.25 \times 10^6, 0.5 \times 10^6, 0.7 \times 10^6$ and $0.9 \times 10^6$ m.

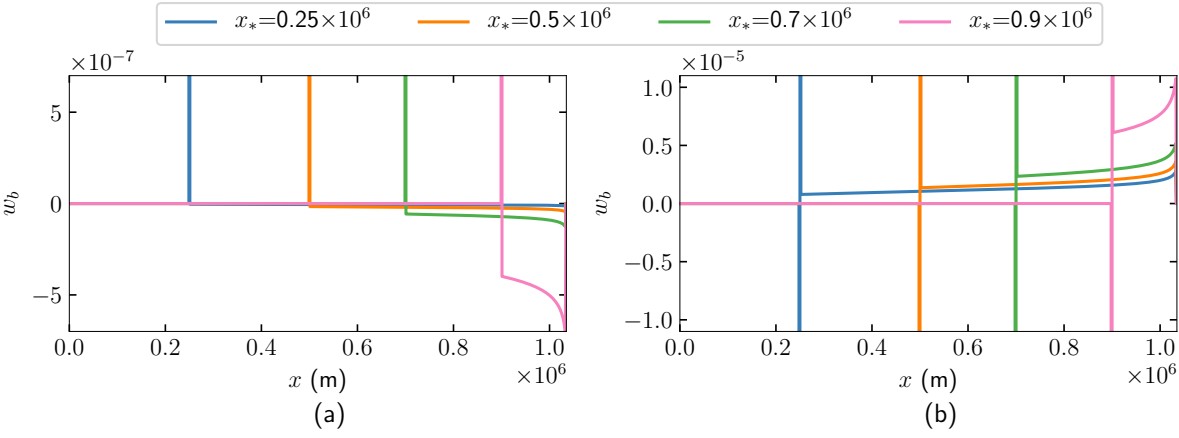

**Figure 7.** The analytical solution of weights $w_b = \psi_x u + v_x \eta u_x + v \rho g h_x$ on $\delta b$ in (31) for (a) $u$ and (b) $h$ observed at $x_* = 0.25 \times 10^6, 0.5 \times 10^6, 0.7 \times 10^6$ and $0.9 \times 10^6$ m.





(iii). When the variation in ice thickness is small compared to the overall ice thickness, $H_x \ll H$, a small perturbation in basal topography $\delta b$ is directly visible in the surface velocity. This is because in such a case, $\delta u_* \approx u_* \delta b_*/H_*$ and the main effect on $u_*$ from the perturbation $\delta b$ is localized at each $x_*$, see (37).

(iv). For an unperturbed basal topography, two different perturbations of the friction coefficient will result in the same pertur-
bation of the velocity. In other words: the perturbation $\delta C$ cannot be uniquely determined by one observation of $\delta u$. This follows if we let the perturbation of the friction coefficient be a constant $\delta C_0 \neq 0$ in $[x_0, x_1] \in [x_*, x_{GL}]$, and evaluate the integral in (37) to obtain

$$\delta u_* = -\frac{u_*}{\rho g H_*^{m+2}} \int_{x_0}^{x_1} (ax)^m \delta C_0 \, \mathrm{d}x = -\frac{a^m u_*}{(m+1)\rho g H_*^{m+2}}(x_1^{m+1} - x_0^{m+1})\delta C_0. \tag{39}$$

The same $\delta u_*$ is observed with a constant perturbation in $[x_2, x_3] \in [x_*, x_{GL}]$ with the amplitude $\delta C_0(x_1^{m+1}-x_0^{m+1})/(x_3^{m+1}-$
$x_2^{m+1})$.

(v). A rapidly varying friction coefficient at the base of the ice sheet will be difficult to identify by observing the velocity at the ice surface. In contrast, a smoothly varying friction coefficient at the base will be easily observable at the ice sheet surface. This is seen as follows: Perturb $C$ by $\delta C = \epsilon \cos(kx/x_{GL})$ in (37) for some wave number $k$ which determines the smoothness of the friction at the bedrock and amplitude $\epsilon$ and let $\delta b = 0$ and $m = 1$. The wavelength is $\lambda = 2\pi x_{GL}/k$.
When $k$ is small then the wavelength is long and the variation of $C + \delta C$ is smooth. When $k$ is large then the friction coefficient varies rapidly in $x$ with a short $\lambda$. The perturbation in the velocity is

$$\begin{aligned}
\delta u_* &= -\int_{x_*}^{x_{GL}} \epsilon \frac{a^2 x_*}{\rho g H_*^4} x \cos\left(\frac{kx}{x_{GL}}\right) \mathrm{d}x \\
&= -\epsilon \frac{a^2 x_*}{\rho g H_*^4} \frac{x_{GL}^2}{k}\left(\sin(k) - \frac{x_*}{x_{GL}}\sin\left(\frac{kx_*}{x_{GL}}\right) + \frac{1}{k}\left(\cos(k) - \cos\left(\frac{kx_*}{x_{GL}}\right)\right)\right).
\end{aligned} \tag{40}$$

For a thin ice with a small $H_*$, a perturbation in $C$ is easier to observe at the surface than for a thick ice. When $k$ grows at the ice base, the amplitude of the perturbation at the ice surface decays as $1/k$. Thus, the effect of high wave number
perturbations of $C$ will be difficult to observe at the top of the ice but smooth perturbations at the base will propagate to the surface. If $k$ is large and the surface velocity is of interest in a numerical simulation, then there is no reason to use a fine mesh at the base to resolve the fast variation in $C$ because it will not be visible at the top of the ice.

(vi). A perturbation in the topography with long wavelength is easier to detect at the surface than a perturbation with short wavelength. If $\delta C = 0$ and $b$ is perturbed by $\delta b = \epsilon \cos(kx/x_{GL})$, then any perturbation at $x_*$ is propagated to the surface
by $\frac{u_* \delta b_*}{H_*}$, which is the first term on the right hand side of (37). The effect is larger if the ice is thin and moving fast. The integral term will behave in the same way as in (40) with mainly perturbations with small wave numbers and long wavelengths visible at the surface.

Finally, let us comment on other approaches to investigate the sensitivity of surface data to changes in $b$ and $C$, e.g. using three linear models as in Gudmundsson (2008) and along a flow line at steady state in Gudmundsson and Raymond (2008)





with a linearized FS model with $n = 1$ and $m = 1$. In these papers, transfer functions for the perturbations from base to surface corresponding to our formulas (37) and (38) are derived by Fourier and Laplace analysis. The perturbations with long wavelength $\lambda$ and small wave number $k$ are propagated to the surface but short wavelengths are effectively damped in Gudmundsson (2008). The transfer functions are utilized in Gudmundsson and Raymond (2008) to estimate how well basal

data can be retrieved from surface data. Retrieval of basal slipperiness $C$ is possible for perturbations $\delta C$ of long wavelength and if the errors in the basal topography $\delta b$ is small. Short wavelength perturbations $\delta b$ can be determined from surface data. The same conclusions as in Gudmundsson (2008) and Gudmundsson and Raymond (2008) can be drawn from our explicit expressions for the dependence of $\delta u_*$ and $\delta h_*$ on $\delta C$ and $\delta b$. For example, it follows from (40) that only $\delta C$ with a long wavelength is visible at the surface and that $\delta b$ also with a short wavelength affects $\delta u_*$ in (38). If $\delta b$ is small or zero in (38),

then it is easier to determine the $\delta C$ that causes a certain $\delta u_*$.

### 3.2.4 The two-dimensional adjoint steady state solution with $F_h \neq 0$.

In the case when $h$ is observed at $x_*$ and $F_u = 0$ and $F_h = \delta(x - x_*)$, the expressions for $\psi$ and $v$ satisfying (34) are

$$
\begin{aligned}
\psi(x) &= -\frac{Ca^{m-1}}{\rho g H_*^{m+1}}\left(x_{GL}^m - x^m\right),\ x_* < x \le x_{GL}, \\
\psi(x) &= -\frac{Ca^{m-1}}{\rho g H_*^{m+1}}\left(x_{GL}^m - x_*^m\right),\ 0 \le x < x_*, \\
v(x) &= -\frac{H^m}{\rho g H_*^{m+1}},\ x_* < x \le x_{GL}, \\
v(x) &= 0,\ 0 \le x < x_*.
\end{aligned}
\tag{41}
$$

The corresponding formulas when $u$ is observed are found in (36). There is a discontinuity at the observation point $x_*$ in $v(x)$

illustrated in Fig. 5(b), but $\psi(x)$ is continuous in the solution of (34) and in Fig. 4(b).

The second derivative term $\left(\frac{1}{n}\eta H v_x\right)_x$ is neglected in the simplified equation (34) but is of importance at $x_*$. A correction $\hat{\psi}$ of $\psi$ at $x_*$ in (41) is therefore introduced to satisfy $\left(\frac{1}{n}\eta H v_x\right)_x - H\hat{\psi}_x = 0$. With $v_x(x_*) = -\delta(x - x_*)/(\rho g H_*)$, the correction is $\hat{\psi}(x) = -\delta(x - x_*)\eta_*/(n\rho g H_*)$. The solution $\psi$ is updated at each $x_*$ in Fig. 4(b) with $\hat{\psi}$ as a vertical line representing the negative Dirac delta.

The perturbation in $h$ is as in (37) with $\psi$ and $v$ in (41) and the additional term $\hat{\psi}$

$$
\begin{aligned}
\frac{\delta h_*}{H_*} &= \int\limits_{x_*^-}^{x_{GL}} -\frac{u\eta_*}{n\rho g H_*^2}\delta_x(x - x_*)\delta b\,\mathrm{d}x + \int\limits_{x_*}^{x_{GL}} \frac{C(ax)^m}{\rho g H_*^{m+2}}\left((m+1)\frac{\delta b}{H} + \frac{\delta C}{C}\right)\,\mathrm{d}x \\
&= \frac{a\eta_*}{n\rho g H_*^2}\left(x\frac{\delta b}{H}\right)_x(x_*) + \frac{1}{\rho g H_*^{m+2}}\int\limits_{x_*}^{x_{GL}} C(ax)^m\left((m+1)\frac{\delta b}{H} + \frac{\delta C}{C}\right)\,\mathrm{d}x,
\end{aligned}
\tag{42}
$$

where $a(x\delta b/H)_x(x_*) = (u\delta b)_x(x_*)$ represents the $x$-derivative of $u\delta b$ evaluated at $x_*$. When $\delta b = 0$ then $\delta u_*$ in (38) and $\delta h_* = \delta H_*$ in (42) satisfy $\delta u_* H_* = -\delta H_* u_*$ as in the integrated form of the advection equation in (33) and in (D1).



As in (37), (42) is rewritten with the weights $w_b$ and $w_C$ in (32)

$$\delta h_* = \int_0^{x_{GL}} (\psi_x u + v_x \eta u_x + v \rho g h_x) \delta b - v u^m \, \delta C \, \mathrm{d}x = \int_0^{x_{GL}} w_b \, \delta b + w_C \, \delta C \, \mathrm{d}x. \tag{43}$$

These weights are shown in Fig. 6(b) and Fig. 7(b). The negative derivative of the Dirac delta is depicted in Fig. 7(b) as a vertical line in the negative direction immediately followed by one in the positive direction.

The contribution from the integrals in (38) and (42) is identical except for the sign (compare $w_C$ in Fig. 6(a) and Fig. 6(b) and $w_b$ in Fig. 7(a) and Fig. 7(b)). The first term in (38) depends on $\delta b / H$ and the first term in (42) depends on the derivative of $a x \delta b / H = u \delta b$. The derivative of $u \delta b$ at $x_*$ directly affects the perturbation of $h$ at $x_*$. A perturbation of $b$ at the base is directly visible locally in $u$ at the surface while the effect of $\delta C$ is non-local in (42). Because of the similarities between (38) and (42) and the left and right columns of Fig. 6 and Fig. 7, the conclusions (i), (ii), (iv), (v), and (vi) in Sect. 3.2.3 from (37)

and (38) for $\delta u_*$ are valid also for $\delta h_*$ in (42).

### 3.2.5    The two-dimensional time dependent adjoint solution.

Finally, the time dependent adjoint equation (30) is investigated. Equation (30) is solved numerically for the same MISMIP test case as in Sect. 3.2.2 with a moving GL. As in Sect. 3.1.1, the friction coefficient $C$ has a seasonal variation (period one year, 1 yr, where the beginning of the year is associated with winter) in the forward equation (29):

$$C(x,t) = C_0(1 + \kappa \cos(2\pi t)), \, 0 < \kappa < 1. \tag{44}$$

Apparently, $C$ has its highest value at $t = n$, $n = 0, 1, 2, \ldots$, i.e. the winter, and its lowest value at $t = n + 1/2$, i.e. the summer, as in Fig. 2. The amplitude of the perturbation is set to $\kappa = 0.5$ and the forward equation (29) is solved for 11 years. The topography $b$ is kept constant in time. Observations of $u$ and $h$ are taken at $x_* = 9 \times 10^5$ m for 0.1 a in the four seasons starting from the summer of the tenth year, e.g., in the summer ($t_* = 9.5$), the fall ($t_* = 9.75$), the winter ($t_* = 10$), and the spring

($t_* = 10.25$). The forward equations (29) are solved numerically from $t = 0$ with the steady state solution as initial data to the observation points $t = t_*$ and the adjoint equations (30) are solved from $t = t_*$ backward in time to $t = 0$. According to a convergence test, the time step is chosen to be 0.01 yr and the spatial resolution is $10^3$ m. A visual inspection of the computed solutions after halving the step sizes indicates that a sufficiently converged solution has been reached.

   Fig. 8 shows the results for the adjoint weights $w_C(x,t)$ and $w_b(x,t)$ multiplying the perturbations $\delta C$ and $\delta b$, as defined in

(31), for the observations of $u$ and $h$ at $x_* = 9 \times 10^5$ m in all four seasons, where each column represents one season. The friction coefficient $C$ follows the seasonal variation in (44). Each row is one of the combinations of the weights $w_C$ and $w_b$ for the observations of $u$ and $h$ . The time axis (or ordinate) in the figure follows the time direction in the forward problem (29). Most of the weights in space and time are negligible implying that perturbations in those domains are not visible at $(x_*, t_*)$. Only $\delta C$ and $\delta b$ in a narrow interval around $x_*$ for $t$ in $[0, t_*]$ have an influence on $\delta u_*$ and $\delta h_*$. Therefore, we take a snapshot

of the $x$ axis (or abscissa) with the width of $10^5$ m in space around $x_*$ in Fig. 8. The weights oscillate in time because of the seasonal variation in the basal conditions. A perturbation at the base is propagated to the $x_*$ position on the surface but with a





possible delay in time. The earlier a perturbation in $C$ or $b$ takes place in the interval $[0, t_*)$, the smaller the effect of it is at $t_*$. After five years a perturbation can hardly be detected at the surface.

The temporal variations of the adjoint weights at $x_*$ in Fig. 8 are shown in Fig. 9 for the four seasons with four different colors. As expected, the weights vanish when $t > t_*$. In Fig. 9(a) and (b), the perturbations $\delta C_*$ and $\delta b_*$ have a direct effect on $\delta u_*$ at

$t_*$, where both $w_C(x_*, t_*)$ and $w_b(x_*, t_*)$ are negative. The same direct effect of $\delta C$ is found for $\delta u_{1*}$ solving the FS equations (17) in Sect. 3.1.1. A change in $\delta C_*$ at the base is observed immediately as a change in $u$ at the surface. The effect of $\delta C$ on $\delta u_*$ for $t < t_*$ is weak in Fig. 9(a), i.e. the memory of old perturbations is short. The largest effect of $\delta C$ on $\delta u_*$ and $\delta h_*$ appears with $t_*$ in the summer when $C$ is small in (44) (the blue lines in Fig. 9(a) and (b)).

However, when $h$ is observed, the effects of $\delta C_*$ and $\delta b_*$ are not visible directly because $w_{C*} \approx 0$ and $w_{b*} \approx 0$ in Fig. 9(c)

and Fig. 9(d). An intuitive explanation is that there is an immediate effect on the velocity but there is a delay in $h$ since it is integrated in time from the velocity field. Additionally, the effects of $\delta C$ and $\delta b$ are difficult to separate, since the weight $w_b(x_*, t)$ has a shape similar to $w_C(x_*, t)$. The largest effect on $\delta h_*$ is from $\delta C$ in the summer due to the peaks in $w_C$ in Fig. 9(c). For the same $\delta C$, the largest $\delta h_*$ is observed in the fall (orange), then the second largest $\delta h_*$ is in the winter (green) followed by the spring observation (red). If $\delta h_*$ is observed in the fall and the time dependency is ignored, then the wrong

conclusion is drawn that $\delta C$ in the fall has the strongest effect (but it is the summer perturbation). There is a delay in time between the perturbation and the observation of the effect in the surface elevation. The same shift in time is what we found in (17) and Fig. 2 for the FS equations.

A reference adjoint solution observed during the fall season ($t_* = 9.75$) with time independent $C$ and $b$, $\kappa = 0$ in (44), in the forward equations is shown in black dashed lines in all the four panels of Fig. 9. The weight $w_b$ for a constant $b$ is well

approximated by $\exp(-(T-t)/\tau)$ in time with $\tau = 1.4$ yr for the observation of both $u$ and $h$. For the weight $w_C$, the same exponential function holds, but the time constant $\tau = 1.8$ yr for the observation of $h_*$ and $\tau = 2.2$ yr for the $u_*$ case.

Suppose that the temporal perturbation is oscillatory $\delta C_0 \cos(2\pi f t)$ with frequency $f$. A low frequency $f$ with $f \ll 1$ corresponds decennial or centennial variations and a high frequency with $f \gg 1$ corresponds to diurnal or weekly variations. Then the perturbation in $h$ at $t = t_*$ is

$$\delta h_* = \int_0^T \exp(-(T-t)/\tau)\delta C_0 \cos(2\pi f t)\, dt = \left( \frac{\cos(2\pi fT) + 2\pi\tau f \sin(2\pi fT) - e^{-T/\tau}}{4\pi^2\tau f^2 + \tau^{-1}} \right)\delta C_0, \qquad (45)$$

cf. (40). With a high frequency, $f \gg 1$, then $\delta h_* \propto 1/f$ and high frequency perturbations are damped efficiently. If the frequency is low, $f \ll 1$, then $\delta h_* \propto \tau$ and the change in $h_*$ is insensitive to the frequency. The same conclusions hold true for $\delta b$ where decennial perturbations seem more realistic.

## 4   Conclusions

The adjoint equations are derived in the FS and the SSA frameworks including time and the surface elevation equation. Time-dependent perturbations $\delta C$ and $\delta b$ in basal friction coefficient $C$ and basal topography $b$ are introduced and their effect on observations of the velocity $u$ and the surface elevation $h$ at the top surface of the ice is studied. The numerical results in Cheng





**Figure 8.** The adjoint weights for the observations at $x_* = 9 \times 10^5$ m of the four seasons. (a) $w_C$ for the observation of $u$. (b) $w_b$ for the observation of $u$. (c) $w_C$ for the observation of $h$. (d) $w_b$ for the observation of $h$.

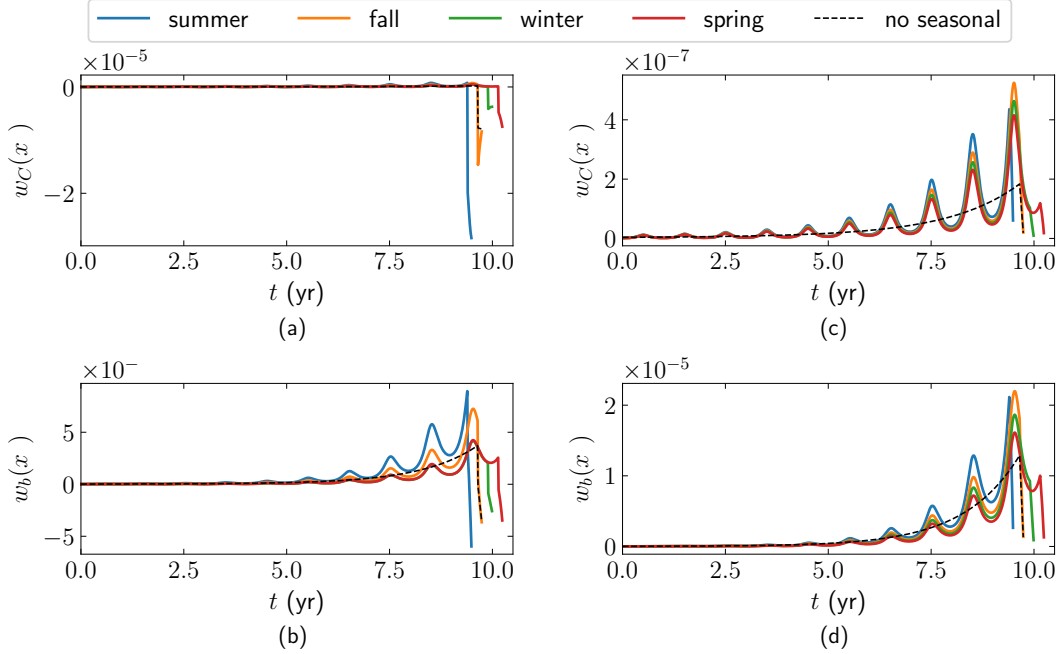

**Figure 9.** The adjoint weights at $x_*$ in the four seasons of the tenth year with seasonally varying friction coefficient. The black dashed line is a reference solution without seasonal variations which is observed at $t_* = 9.75$. (a) $w_C$ for the observation of $u$. (b) $w_b$ for the observation of $u$. (c) $w_C$ for the observation of $h$. (d) $w_b$ for the observation of $h$ .

and Lötstedt (2020) confirm the conclusions here and are in good agreement with the analytical solutions. In Sect. 3.1.2, a relation is established between the inverse problem (aiming to infer parameters from data) and the sensitivity problem (aiming to quantify the effect of variations in parameters): The same adjoint equations are solved. However, the forcing functions differ and are specific to the inverse problem, and the sensitivity problem, respectively. Common to both problems is that the adjoint

equations tell how uncertainties in the parameters at the ice base are propagated to uncertainties in the velocity and the elevation of the surface.

For steady state problems, and in an FS setting where $\boldsymbol{u}$ is observed, we find (cf. Sect. 3.1.3) that the contribution of the solution of the adjoint elevation equation (24) is small, and that it therefore suffices to solve only the adjoint stress equations, see e.g. Gillet-Chaulet et al. (2016); Isaac et al. (2015); Petra et al. (2012), in order to be able to draw conclusions regarding

perturbations of $\boldsymbol{u}$. For steady state problems in a two-dimensional SSA setting, (38), (42), and Figs. 6 and 7 show that the sensitivity of the velocity and elevation increases (because the velocity increases and the ice thickness decreases) as the observation point $x_*$ approaches the grounding line.

In this setting, there is moreover observed a non-local effect of a perturbation in $C$, in the sense that $\delta C(x)$ affects both $u(x_*)$ and $h(x_*)$ even if $x \neq x_*$, but a perturbation $\delta b$ in $b$ has a strong local effect concentrated at $x_*$. Nevertheless, the shapes of

the two sensitivity functions (or, weights) for $\delta b$ and $\delta C$ are very similar except for the neighborhood of $x_*$, which makes it





difficult to separate their respective contribution in an observation. Different combinations of the perturbations in the basal friction and bedrock elevation can produce the same effect on the velocity and surface elevation changes at one observation point.

In the inverse problems based on time dependent simulations of FS and SSA, it is necessary to include the adjoint elevation
equation. If the perturbations in the basal conditions are time dependent and $h$ is observed (see Fig. 2, Fig. 9(c), and Fig. 9(d)), then time cannot be ignored in the inversion. If time dependence *is* ignored, wrong conclusions concerning the conditions at the ice base may be drawn from observations of $h$, in both the FS and the SSA model. In the time dependent solution of SSA, a perturbation of the basal condition at $x_*$ has the strongest impact at $x_*$ on the surface, possibly with a time delay. Such a time delay occurs when a perturbation at the ice base is visible at the surface in $h$, but in $u$ it is observed immediately (Fig. 9). The
effect of a perturbation disappears more quickly, the older the perturbation is.

Perturbations in the friction coefficient at the base observed in the surface velocity determined by SSA are damped inversely proportional to the wave number and the frequency of the perturbations in (40) and (45), thus making very oscillatory perturbations in space and time difficult to register at the ice sheet surface. In such a case, there is no need to have a fine mesh and a small time-step in a numerical solution to resolve the rapid oscillations in $C$ at the base.

**Appendix A: Derivation of the adjoint equations**

**A1 Adjoint viscosity and friction in SSA**

The adjoint viscosity $\tilde{\boldsymbol{\eta}}(\boldsymbol{u})$ in SSA in (14) is derived as follows. The SSA viscosity for $\boldsymbol{u}$ and $\boldsymbol{u} + \delta\boldsymbol{u}$ is

$$
\begin{aligned}
&\eta(\boldsymbol{u} + \delta\boldsymbol{u}) \\
&\approx \eta(\boldsymbol{u}) \left(1 + \frac{1-n}{2n} \frac{(2u_{1x}+u_{2y})\delta u_{1x} + \frac{1}{2}(u_{1y}+u_{2x})\delta u_{2x} + (2u_{2y}+u_{1x})\delta u_{2y} + \frac{1}{2}(u_{1y}+u_{2x})\delta u_{1y}}{\hat{\eta}}\right).
\end{aligned}
\tag{A1}
$$

Determine $\mathcal{B}(\boldsymbol{u})$ such that

$$\varrho(\boldsymbol{u}, \delta\boldsymbol{u})\mathbf{B}(\boldsymbol{u}) = \mathcal{B}(\boldsymbol{u}) \star \mathbf{B}(\delta\boldsymbol{u}).$$

First note that

$$
\begin{aligned}
&\mathbf{B}(\boldsymbol{u}) : \mathbf{D}(\delta\boldsymbol{u}) = (\mathbf{D}(\boldsymbol{u}) + \nabla\cdot\boldsymbol{u}\mathbf{I}) : \mathbf{D}(\delta\boldsymbol{u}) = \mathbf{D}(\boldsymbol{u}) : \mathbf{D}(\delta\boldsymbol{u}) + (\nabla\cdot\boldsymbol{u})(\nabla\cdot\delta\boldsymbol{u}) \\
&= \mathbf{D}(\boldsymbol{u}) : (\mathbf{B}(\delta\boldsymbol{u}) - \nabla\cdot\delta\boldsymbol{u}\mathbf{I}) + (\nabla\cdot\boldsymbol{u})(\nabla\cdot\delta\boldsymbol{u}) = \mathbf{D}(\boldsymbol{u}) : \mathbf{B}(\delta\boldsymbol{u}).
\end{aligned}
$$

Then use the $\star$ operator to define $\mathcal{B}$

$$
\begin{aligned}
\frac{1-n}{2n\hat{\eta}} \sum_{kl} B_{kl}(\boldsymbol{u}) D_{kl}(\delta\boldsymbol{u}) B_{ij}(\boldsymbol{u}) &= \frac{1-n}{2n\hat{\eta}} \sum_{kl} D_{kl}(\boldsymbol{u}) B_{kl}(\delta\boldsymbol{u}) B_{ij}(\boldsymbol{u}) \\
&= \sum_{kl} \mathcal{B}_{ijkl}(\boldsymbol{u}) D_{kl}(\delta\boldsymbol{u}) = (\mathcal{B} \star D)_{ij}.
\end{aligned}
$$

Thus, let

$$
\mathcal{B}_{ijkl} = \frac{1-n}{2n\hat{\eta}} B_{ij}(\boldsymbol{u}) D_{kl}(\boldsymbol{u}), \quad \tilde{\eta}_{ijkl}(\boldsymbol{u}) = \eta(\boldsymbol{u})(\mathcal{I}_{ijkl} + \mathcal{B}_{ijkl}(\boldsymbol{u})),
$$





or in tensor form

$$\mathcal{B} = \frac{1-n}{n\mathbf{B}(\boldsymbol{u}):\mathbf{D}(\boldsymbol{u})}\mathbf{B}(\boldsymbol{u})\otimes\mathbf{D}(\boldsymbol{u}), \quad \tilde{\boldsymbol{\eta}}(\boldsymbol{u}) = \eta(\boldsymbol{u})\left(\mathcal{I}+\mathcal{B}\right). \tag{A2}$$

Replacing $\mathbf{B}$ in (A2) by $\mathbf{D}$ we obtain the adjoint FS viscosity in (14).

The adjoint friction in SSA in $\omega$ and at $\gamma_g$ in (26) with a Weertman law is derived as in the adjoint FS equations (13) and
(14). Then in $\omega$ with $\boldsymbol{\xi}=\boldsymbol{u}, \boldsymbol{\zeta}=\boldsymbol{v}, c=C, \mathbf{F}=\mathbf{F}_\omega$, and at $\gamma_g$ with $\boldsymbol{\xi}=\boldsymbol{t}\cdot\boldsymbol{u}, \boldsymbol{\zeta}=\boldsymbol{t}\cdot\boldsymbol{v}, c=C_\gamma, f=f_\gamma, \mathbf{F}=F_\gamma$, we arrive at the
adjoint friction term $cf(\boldsymbol{\xi})\left(\mathbf{I}+\mathbf{F}(\boldsymbol{\xi})\right)\boldsymbol{\zeta}$ where

$$\mathbf{F}(\boldsymbol{\xi}) = \frac{m-1}{\boldsymbol{\xi}\cdot\boldsymbol{\xi}}\boldsymbol{\xi}\otimes\boldsymbol{\xi}. \tag{A3}$$

## A2    Adjoint equations in SSA

The Lagrangian for the SSA equations is with the adjoint variables $\psi, \boldsymbol{v}, q$

$$\begin{aligned}
\mathcal{L}(\boldsymbol{u},h;\boldsymbol{v},\psi;b,C_\gamma,C) &= \int_0^T\int_\omega F(\boldsymbol{u},h)+\psi(h_t+\nabla\cdot(\boldsymbol{u}H)-a)\,\mathrm{d}\boldsymbol{x}\,\mathrm{d}t \\
&+\int_0^T\int_\omega \boldsymbol{v}\cdot\nabla\cdot(2H\eta\mathbf{B}(\boldsymbol{u}))-Cf(\boldsymbol{u})\boldsymbol{v}\cdot\boldsymbol{u}-\rho gH\boldsymbol{v}\cdot\nabla h\,\mathrm{d}\boldsymbol{x}\,\mathrm{d}t \\
&= \int_0^T\int_\omega F(\boldsymbol{u},h)+\psi(h_t+\nabla\cdot(\boldsymbol{u}H)-a)\,\mathrm{d}\boldsymbol{x}\,\mathrm{d}t \\
&+\int_0^T\int_\omega -2H\eta(\boldsymbol{u})(\mathbf{D}(\boldsymbol{v}):\mathbf{D}(\boldsymbol{u})+\nabla\cdot\boldsymbol{u}\nabla\cdot\boldsymbol{v}) \\
&-Cf(\boldsymbol{u})\boldsymbol{v}\cdot\boldsymbol{u}-\rho gH\boldsymbol{v}\cdot\nabla h\,\mathrm{d}\boldsymbol{x}\,\mathrm{d}t-\int_0^T\int_{\gamma_g}C_\gamma f_\gamma(\boldsymbol{t}\cdot\boldsymbol{u})\boldsymbol{t}\cdot\boldsymbol{u}\boldsymbol{t}\cdot\boldsymbol{v}\,\mathrm{d}s\,\mathrm{d}t
\end{aligned} \tag{A4}$$

after partial integration and using the boundary conditions. The perturbed SSA Lagrangian is split into the unperturbed Lagrangian and three integrals

$$\begin{aligned}
&\mathcal{L}(\boldsymbol{u}+\delta\boldsymbol{u},h+\delta h;\boldsymbol{v}+\delta\boldsymbol{v},\psi+\delta\psi;b+\delta b,C_\gamma+\delta C_\gamma,C+\delta C) \\
&= \int_0^T\int_\omega F(\boldsymbol{u}+\delta\boldsymbol{u},h+\delta h) \\
&+\int_0^T\int_\omega(\psi+\delta\psi)(h_t+\delta h_t+\nabla\cdot((\boldsymbol{u}+\delta\boldsymbol{u})(H+\delta H))-a)\,\mathrm{d}\boldsymbol{x}\,\mathrm{d}t \\
&+\int_0^T\int_\omega -2(H+\delta H)\eta(\boldsymbol{u}+\delta\boldsymbol{u})\mathbf{D}(\boldsymbol{v}+\delta\boldsymbol{v}):\mathbf{B}(\boldsymbol{u}+\delta\boldsymbol{u}) \\
&-(C+\delta C)f(\boldsymbol{u}+\delta\boldsymbol{u})(\boldsymbol{u}+\delta\boldsymbol{u})\cdot(\boldsymbol{v}+\delta\boldsymbol{v}) \\
&-\rho g(H+\delta H)\nabla(h+\delta h)\cdot(\boldsymbol{v}+\delta\boldsymbol{v})\,\mathrm{d}\boldsymbol{x}\,\mathrm{d}t \\
&-\int_0^T\int_{\gamma_g}(C_\gamma+\delta C_\gamma)f_\gamma(\boldsymbol{t}\cdot(\boldsymbol{u}+\delta\boldsymbol{u}))\boldsymbol{t}\cdot(\boldsymbol{u}+\delta\boldsymbol{u})\boldsymbol{t}\cdot(\boldsymbol{v}+\delta\boldsymbol{v})\,\mathrm{d}s\,\mathrm{d}t \\
&= \mathcal{L}(\boldsymbol{u},h;\boldsymbol{v},\psi;b,C_\gamma,C)+I_1+I_2+I_3.
\end{aligned} \tag{A5}$$

The perturbation in $\mathcal{L}$ is

$$\delta\mathcal{L} = I_1+I_2+I_3. \tag{A6}$$

Terms of order two or more in $\delta\mathcal{L}$ are neglected. Then the first term in $\delta\mathcal{L}$ satisfies

$$\begin{aligned}
I_1 &= \int_0^T\int_\omega F(\boldsymbol{u}+\delta\boldsymbol{u},h+\delta h)-F(\boldsymbol{u},h)\,\mathrm{d}\boldsymbol{x}\,\mathrm{d}t \\
&= \int_0^T\int_\omega F_{\boldsymbol{u}}\delta\boldsymbol{u}+F_h\delta h\,\mathrm{d}\boldsymbol{x}\,\mathrm{d}t.
\end{aligned} \tag{A7}$$





Using partial integration, Gauss' formula, and the initial and boundary conditions on $\boldsymbol{u}$ and $H$ and $\psi(\boldsymbol{x}, T) = 0, \boldsymbol{x} \in \omega$, and $\psi(\boldsymbol{x}, t) = 0$, $\boldsymbol{x} \in \gamma_w$, in the second integral we have

$$
\begin{aligned}
I_2 &= \int_0^T \int_\omega \delta\psi(h_t + \nabla \cdot (\boldsymbol{u}H) - a) \\
&\quad + \psi(\delta h_t + \nabla \cdot (\delta\boldsymbol{u}H) + \nabla \cdot (\boldsymbol{u}\delta H)) \, d\boldsymbol{x} \, dt \\
&= \int_0^T \int_\omega \delta\psi(h_t + \nabla \cdot (\boldsymbol{u}H) - a) \, d\boldsymbol{x} \, dt \\
&\quad + \int_0^T \int_\omega -\psi_t \delta h - H \nabla\psi \cdot \delta\boldsymbol{u} - \nabla\psi \cdot \boldsymbol{u} \delta H \, d\boldsymbol{x} \, dt.
\end{aligned}
\tag{A8}
$$

The first integral after the second equality vanishes since $h$ is a weak solution and $I_2$ is

$$
I_2 = \int_0^T \int_\omega -(\psi_t + \boldsymbol{u} \cdot \nabla\psi)\delta h - H \nabla\psi \cdot \delta\boldsymbol{u} + \boldsymbol{u} \cdot \nabla\psi \delta b \, d\boldsymbol{x} \, dt.
\tag{A9}
$$

Using the weak solution of (8), the adjoint viscosity (27), (A2), the friction coefficient (A3), Gauss' formula, the boundary conditions, and neglecting the second order terms, the third and fourth integrals in (A5) are

$$
\begin{aligned}
I_3 &= I_{31} + I_{32}, \\
I_{31} &= \int_0^T \int_\omega -2(H + \delta H)\eta(\boldsymbol{u} + \delta\boldsymbol{u})\mathbf{D}(\boldsymbol{v} + \delta\boldsymbol{v}) : \mathbf{B}(\boldsymbol{u} + \delta\boldsymbol{u}) \\
&\quad -(C + \delta C)f(\boldsymbol{u} + \delta\boldsymbol{u})(\boldsymbol{u} + \delta\boldsymbol{u}) \cdot (\boldsymbol{v} + \delta\boldsymbol{v}) \\
&\quad -\rho g(H + \delta H)\nabla(h + \delta h) \cdot (\boldsymbol{v} + \delta\boldsymbol{v})) \, d\boldsymbol{x} \, dt \\
&\quad - \int_0^T \int_\gamma (C_\gamma + \delta C_\gamma)f_\gamma(\boldsymbol{t} \cdot (\boldsymbol{u} + \delta\boldsymbol{u}))\boldsymbol{t} \cdot (\boldsymbol{u} + \delta\boldsymbol{u})\boldsymbol{t} \cdot (\boldsymbol{v} + \delta\boldsymbol{v}) \, ds \, dt \\
&= I_{311} + I_{312} - I_{313},
\end{aligned}
\tag{A10}
$$

where

$$
\begin{aligned}
I_{311} &= \int_0^T \int_\omega -2H\mathbf{D}(\boldsymbol{v}) : (\eta(\boldsymbol{u} + \delta\boldsymbol{u})\mathbf{B}(\boldsymbol{u} + \delta\boldsymbol{u})) \\
&\quad +2H\mathbf{D}(\boldsymbol{v}) : (\eta(\boldsymbol{u})\mathbf{B}(\boldsymbol{u})) \, d\boldsymbol{x} \, dt \\
&= \int_0^T \int_\omega -2H\mathbf{D}(\boldsymbol{v}) : (\tilde{\boldsymbol{\eta}}(\boldsymbol{u}) \star \mathbf{B}(\delta\boldsymbol{u})) \, d\boldsymbol{x} \, dt \\
I_{312} &= \int_0^T \int_{\omega_g} -\delta C f(\boldsymbol{u})\boldsymbol{u} \cdot \boldsymbol{v} \, d\boldsymbol{x} \, dt \\
&\quad + \int_0^T \int_{\omega_g} -C(f(\boldsymbol{u} + \delta\boldsymbol{u})\boldsymbol{v} \cdot (\boldsymbol{u} + \delta\boldsymbol{u}) - f(\boldsymbol{u})\boldsymbol{v} \cdot \boldsymbol{u}) \, d\boldsymbol{x} \, dt \\
&= \int_0^T \int_{\omega_g} -\delta C f(\boldsymbol{u})\boldsymbol{u} \cdot \boldsymbol{v} + C f(\boldsymbol{u})(\mathbf{I} + \mathbf{F}_\omega(\boldsymbol{u}))\delta\boldsymbol{u} \cdot \boldsymbol{v} \, d\boldsymbol{x} \, dt \\
I_{313} &= \int_0^T \int_{\gamma_g} (C_\gamma + \delta C_\gamma)(f_\gamma(\boldsymbol{t} \cdot (\boldsymbol{u} + \delta\boldsymbol{u}))\boldsymbol{t} \cdot \boldsymbol{v}\boldsymbol{t} \cdot (\boldsymbol{u} + \delta\boldsymbol{u}) \\
&\quad -f_\gamma(\boldsymbol{t} \cdot \boldsymbol{u})\boldsymbol{t} \cdot \boldsymbol{v}\boldsymbol{t} \cdot \boldsymbol{u}) \, ds \, dt \\
&= \int_0^T \int_{\gamma_g} (C_\gamma + \delta C_\gamma)(f_\gamma(\boldsymbol{t} \cdot \boldsymbol{u})\boldsymbol{t} \cdot \boldsymbol{u}\boldsymbol{t} \cdot \boldsymbol{v} \\
&\quad + C_\gamma f_\gamma(\boldsymbol{t} \cdot \boldsymbol{u})(\mathbf{I} + \mathbf{F}_\gamma(\boldsymbol{t} \cdot \boldsymbol{u}))\boldsymbol{t} \cdot \delta\boldsymbol{u}\boldsymbol{t} \cdot \boldsymbol{v} \, ds \, dt \\
I_{32} &= \int_0^T \int_\omega -\rho g H \nabla h \cdot \boldsymbol{v} - 2\eta\mathbf{D}(\boldsymbol{v}) : \mathbf{B}(\boldsymbol{u})\delta H \\
&\quad -\rho g \nabla h \cdot \boldsymbol{v}\delta H - \rho g H\boldsymbol{v} \cdot \nabla\delta h \, d\boldsymbol{x} \, dt \\
&= \int_0^T \int_\omega -\rho g H \nabla h \cdot \boldsymbol{v} - (2\eta\mathbf{D}(\boldsymbol{v}) : \mathbf{B}(\boldsymbol{u}) + \rho g \nabla h \cdot \boldsymbol{v})\delta H \\
&\quad + \rho g \nabla \cdot (H\boldsymbol{v})\delta h \, d\boldsymbol{x} \, dt.
\end{aligned}
\tag{A11}
$$





Collecting all the terms in (A7), (A9), and (A10), the first variation of $\mathcal{L}$ is

$$
\begin{aligned}
\delta\mathcal{L} &= I_1 + I_2 + I_3 \\
&= \int_0^T \int_\omega F_{\boldsymbol{u}} \delta\boldsymbol{u} - 2HD(\boldsymbol{v}) : (\tilde{\boldsymbol{\eta}}(\boldsymbol{u}) \star \mathbf{B}(\delta\boldsymbol{u})) - H\nabla\psi \cdot \delta\boldsymbol{u}\,\mathrm{d}\boldsymbol{x}\,\mathrm{d}t \\
&\quad - \int_0^T \int_{\omega_g} Cf(\boldsymbol{u})(\mathbf{I} + \mathbf{F}_\omega(\boldsymbol{u}))\boldsymbol{v} \cdot \delta\boldsymbol{u}\,\mathrm{d}\boldsymbol{x}\,\mathrm{d}t \\
&\quad - \int_0^T \int_{\gamma_g} C_\gamma f_\gamma(\boldsymbol{t}\cdot\boldsymbol{u})(\mathbf{I} + \mathbf{F}_\gamma(\boldsymbol{t}\cdot\boldsymbol{u}))\boldsymbol{t}\cdot\boldsymbol{v}\boldsymbol{t}\cdot\delta\boldsymbol{u}\,\mathrm{d}s\,\mathrm{d}t \\
&\quad - \int_0^T \int_{\gamma_g} \delta C_\gamma f_\gamma(\boldsymbol{t}\cdot\boldsymbol{u})\boldsymbol{t}\cdot\boldsymbol{u}\boldsymbol{t}\cdot\boldsymbol{v}\,\mathrm{d}s\,\mathrm{d}t \\
&\quad + \int_0^T \int_\omega (F_h - (\psi_t + \boldsymbol{u}\cdot\nabla\psi + 2\eta\mathbf{D}(\boldsymbol{v}) : \mathbf{B}(\boldsymbol{u}) \\
&\quad - \rho g\nabla b\cdot\boldsymbol{v} + \rho gH\nabla\cdot\boldsymbol{v}))\delta h\,\mathrm{d}\boldsymbol{x}\,\mathrm{d}t \\
&\quad + \int_0^T \int_\omega -\delta Cf(\boldsymbol{u})\boldsymbol{v}\cdot\boldsymbol{u} \\
&\quad + (2\eta\mathbf{D}(\boldsymbol{v}) : \mathbf{B}(\boldsymbol{u}) + \rho g\nabla h\cdot\boldsymbol{v} + \boldsymbol{u}\cdot\nabla\psi)\delta b\,\mathrm{d}\boldsymbol{x}\,\mathrm{d}t.
\end{aligned}
\tag{A12}
$$

The forward solution $(\boldsymbol{u}^*, p^*, h^*)$ and adjoint solution $(\boldsymbol{v}^*, q^*, \psi^*)$ satisfying (8) and (26) are inserted into (A4) resulting in

$$
\mathcal{L}(\boldsymbol{u}^*, p^*; \boldsymbol{v}^*, q^*; h^*, \psi^*; b, C_\gamma, C) = \int_0^T \int_\omega F(\boldsymbol{u}^*, h^*)\,\mathrm{d}\boldsymbol{x}\,\mathrm{d}t.
\tag{A13}
$$

5  Then (A12) yields the variation in $\mathcal{L}$ in (A13) with respect to perturbations $\delta b, \delta C_\gamma$, and $\delta C$ in $b, C_\gamma$, and $C$

$$
\begin{aligned}
\delta\mathcal{L} &= \int_0^T \int_\omega (2\eta\mathbf{D}(\boldsymbol{v}^*) : \mathbf{B}(\boldsymbol{u}^*) + \rho g\nabla h^* \cdot \boldsymbol{v}^* + \boldsymbol{u}^* \cdot \nabla\psi^*)\delta b\,\mathrm{d}\boldsymbol{x}\,\mathrm{d}t \\
&\quad - \int_0^T \int_{\gamma_g} \delta C_\gamma f_\gamma(\boldsymbol{t}\cdot\boldsymbol{u}^*)\boldsymbol{t}\cdot\boldsymbol{u}^*\boldsymbol{t}\cdot\boldsymbol{v}^*\,\mathrm{d}s\,\mathrm{d}t \\
&\quad - \int_0^T \int_\omega \delta Cf(\boldsymbol{u}^*)\boldsymbol{v}^* \cdot \boldsymbol{u}^*\,\mathrm{d}\boldsymbol{x}\,\mathrm{d}t.
\end{aligned}
\tag{A14}
$$

## A3  Adjoint equations in FS

The FS Lagrangian is

$$
\begin{aligned}
\mathcal{L}(\boldsymbol{u}, p, h; \boldsymbol{v}, q, \psi; C) &= \int_0^T \int_{\Gamma_s} F(\boldsymbol{u}, h) + \psi(h_t + \boldsymbol{h}\cdot\boldsymbol{u} - a)\,dx\,\mathrm{d}t \\
&\quad + \int_0^T \int_\omega \int_b^h -\boldsymbol{v}\cdot(\nabla\cdot\boldsymbol{\sigma}(\boldsymbol{u}, p)) - q\nabla\cdot\boldsymbol{u} - \rho\boldsymbol{g}\cdot\boldsymbol{v}\,\mathrm{d}\boldsymbol{x}\,\mathrm{d}t \\
&= \int_0^T \int_{\Gamma_s} F(\boldsymbol{u}, h) + \psi(h_t + \boldsymbol{h}\cdot\boldsymbol{u} - a)\,\mathrm{d}\boldsymbol{x}\,\mathrm{d}t \\
&\quad + \int_0^T \int_\omega \int_b^h 2\eta(\boldsymbol{u})\mathbf{D}(\boldsymbol{v}) : \mathbf{D}(\boldsymbol{u}) - p\nabla\cdot\boldsymbol{v} - q\nabla\cdot\boldsymbol{u} - \rho\boldsymbol{g}\cdot\boldsymbol{v}\,\mathrm{d}\boldsymbol{x}\,\mathrm{d}t \\
&\quad + \int_0^T \int_{\Gamma_b} Cf(\mathbf{T}\boldsymbol{u})\mathbf{T}\boldsymbol{u}\cdot\mathbf{T}\boldsymbol{v}\,\mathrm{d}\boldsymbol{x}\,\mathrm{d}t.
\end{aligned}
\tag{A15}
$$

10  In the same manner as in (A5), the perturbed FS Lagrangian is

$$
\begin{aligned}
&\mathcal{L}(\boldsymbol{u} + \delta\boldsymbol{u}, p + \delta p; \boldsymbol{v} + \delta\boldsymbol{v}, q + \delta q; h + \delta h, \psi + \delta\psi; C + \delta C) \\
&= \mathcal{L}(\boldsymbol{u}, p, h; \boldsymbol{v}, q, \psi; C) + I_1 + I_2 + I_3.
\end{aligned}
\tag{A16}
$$

Terms of order two or more in $\delta\boldsymbol{u}, \delta\boldsymbol{v}, \delta h$ are neglected. The first integral $I_1$ in (A16) is

$$
\begin{aligned}
I_1 &= \int_0^T \int_{\Gamma_s} F(\boldsymbol{u}(\boldsymbol{x}, h + \delta h, t) + \delta\boldsymbol{u}, h + \delta h) - F(\boldsymbol{u}(\boldsymbol{x}, h, t), h)\,\mathrm{d}\boldsymbol{x}\,\mathrm{d}t \\
&= \int_0^T \int_{\Gamma_s} F_{\boldsymbol{u}}(\delta\boldsymbol{u} + \boldsymbol{u}_z\delta h) + F_h\delta h\,\mathrm{d}\boldsymbol{x}\,\mathrm{d}t.
\end{aligned}
\tag{A17}
$$





Partial integration, the conditions $\psi(\boldsymbol{x},T)=0$ and $\psi(\boldsymbol{x},t)=0$ at $\Gamma_s$, and the fact that $h$ is a weak solution simplify the second integral

$$
\begin{aligned}
I_2 &= \int_0^T \int_{\Gamma_s} \delta\psi(h_t + \boldsymbol{h}\cdot\boldsymbol{u} - a) \\
&\quad + \psi(\delta h_t + \boldsymbol{u}\cdot\delta\boldsymbol{h} + \boldsymbol{u}_z\cdot\boldsymbol{h}\delta h + \boldsymbol{h}\cdot\delta\boldsymbol{u})\,\mathrm{d}\boldsymbol{x}\,\mathrm{d}t \\
&= \int_0^T \int_{\Gamma_s} \delta\psi(h_t + \boldsymbol{h}\cdot\boldsymbol{u} - a)\,\mathrm{d}\boldsymbol{x}\,\mathrm{d}t \\
&\quad + \int_0^T \int_{\Gamma_s} (-\psi_t - \nabla\cdot(\boldsymbol{u}\psi) + \boldsymbol{h}\cdot\boldsymbol{u}_z\psi)\delta h + \boldsymbol{h}\cdot\delta\boldsymbol{u}\psi\,\mathrm{d}\boldsymbol{x}\,\mathrm{d}t.
\end{aligned}
\tag{A18}
$$

Define $\Xi,\xi,$ and $\Upsilon$ to be

$$
\begin{aligned}
\Theta(\boldsymbol{u},p;\boldsymbol{v},q;C) &= 2\eta(\boldsymbol{u})\mathbf{D}(\boldsymbol{v}):\mathbf{D}(\boldsymbol{u}) - p\nabla\cdot\boldsymbol{v} - q\nabla\cdot\boldsymbol{u} - \rho\boldsymbol{g}\cdot\boldsymbol{v}, \\
\theta(\boldsymbol{u};\boldsymbol{v};C) &= Cf(\mathbf{T}\boldsymbol{u})\mathbf{T}\boldsymbol{u}\cdot\mathbf{T}\boldsymbol{v}, \\
\Upsilon(\boldsymbol{u},p;\boldsymbol{v},q) &= -\boldsymbol{v}\cdot(\nabla\cdot\boldsymbol{\sigma}(\boldsymbol{u},p)) - q\nabla\cdot\boldsymbol{u} - \rho\boldsymbol{g}\cdot\boldsymbol{v}.
\end{aligned}
\tag{A19}
$$

Then a weak solution, $(\boldsymbol{u},p)$, for any $(\boldsymbol{v},q)$ satisfying the boundary conditions, fulfills

$$
\int_0^T \int_\omega \int_b^h \Theta(\boldsymbol{u},p;\boldsymbol{v},q;C)\,\mathrm{d}\boldsymbol{x}\,\mathrm{d}t - \int_0^T \int_{\Gamma_b} \theta(\boldsymbol{u};\boldsymbol{v};C)\,\mathrm{d}\boldsymbol{x}\,\mathrm{d}t = 0.
\tag{A20}
$$

The third integral in (A16) is

$$
\begin{aligned}
I_3 &= I_{31} + I_{32}, \\
I_{31} &= \int_0^T \int_\omega \int_b^h \Theta(\boldsymbol{u}+\delta\boldsymbol{u},p+\delta p;\boldsymbol{v}+\delta\boldsymbol{v},q+\delta q;C+\delta C)\,\mathrm{d}\boldsymbol{x}\,\mathrm{d}t \\
&\quad - \int_0^T \int_{\Gamma_b} \theta(\boldsymbol{u}+\delta\boldsymbol{u};\boldsymbol{v}+\delta\boldsymbol{v};C+\delta C)\,\mathrm{d}\boldsymbol{x}\,\mathrm{d}t, \\
I_{32} &= \int_0^T \int_\omega \int_h^{h+\delta h} \Upsilon(\boldsymbol{u},p;\boldsymbol{v},q)\,\mathrm{d}\boldsymbol{x}\,\mathrm{d}t.
\end{aligned}
\tag{A21}
$$

10   The integral $I_{31}$ is expanded as in (A10) and (A11) or Petra et al. (2012) using the weak solution, Gauss' formula, and the definitions of the adjoint viscosity and adjoint friction coefficient in Appendix A1. When $b < z < h$ we have $\Upsilon(\boldsymbol{u},p;\boldsymbol{v},q) = 0$. If $\Upsilon$ is extended smoothly in the positive $z$-direction from $z = h$, then with $z \in [h, h+\delta h]$ for some constant $c > 0$ we have $|\Upsilon| \leq c\delta h$. Therefore,

$$
\left| \int_h^{h+\delta h(x,t)} \Upsilon(\boldsymbol{u},p;\boldsymbol{v},q)\,dz \right| \leq \int_h^{h+\delta h(x,t)} \sup|\Upsilon|\,dz \leq c|\delta h(x,t)^2|,
$$

15   and the bound on $I_{32}$ in (A21) is

$$
|I_{32}| \leq ct|\omega|\max|\delta h(x,t)|^2,
\tag{A22}
$$

where $|\omega|$ is the area of $\omega$. This term is a second variation in $\delta h$ which is neglected and $I_3 = I_{31}$.





The first variation of $\mathcal{L}$ is then

$$
\begin{aligned}
\delta\mathcal{L} &= I_1 + I_2 + I_3 \\
&= \int_0^T \int_{\Gamma_s} (F_{\boldsymbol{u}} + \psi\boldsymbol{h}) \cdot \delta\boldsymbol{u} \, \mathrm{d}\boldsymbol{x} \, \mathrm{d}t \\
&\quad + \int_0^T \int_{\Gamma_s} (F_h + F_{\boldsymbol{u}}\boldsymbol{u}_z - (\psi_t + \nabla\cdot(\boldsymbol{u}\psi) - \boldsymbol{h}\cdot\boldsymbol{u}\psi))\delta h \, \mathrm{d}\boldsymbol{x} \, \mathrm{d}t \\
&\quad + \int_0^T \int_\omega \int_b^h 2\mathbf{D}(\boldsymbol{v}) : (\tilde{\boldsymbol{\eta}}(\boldsymbol{u}) \star \mathbf{D}(\delta\boldsymbol{u})) - \delta p\nabla\cdot\boldsymbol{v} - q\nabla\cdot\delta\boldsymbol{u} \, \mathrm{d}\boldsymbol{x} \, \mathrm{d}t \\
&\quad + \int_0^T \int_{\Gamma_b} Cf(\mathbf{T}\boldsymbol{u})(\mathbf{I} + \mathbf{F}_b(\boldsymbol{u}))\mathbf{T}\boldsymbol{v}\cdot\mathbf{T}\delta\boldsymbol{u} \, \mathrm{d}\boldsymbol{x} \, \mathrm{d}t \\
&\quad + \int_0^T \int_{\Gamma_b} \delta Cf(\mathbf{T}\boldsymbol{u})\mathbf{T}\boldsymbol{u}\cdot\mathbf{T}\boldsymbol{v} \, \mathrm{d}\boldsymbol{x} \, \mathrm{d}t.
\end{aligned}
\tag{A23}
$$

With the forward solution $(\boldsymbol{u}^*, p^*, h^*)$ and the adjoint solution $(\boldsymbol{v}^*, q^*, \psi^*)$ satisfying (4) and (13), the first variation with respect to perturbations $\delta C$ in $C$ is (cf. (A14))

$$
\quad \delta\mathcal{L} = \int_0^T \int_{\Gamma_b} f(\mathbf{T}\boldsymbol{u}^*)\mathbf{T}\boldsymbol{u}^* \cdot \mathbf{T}\boldsymbol{v}^* \, \delta C \, \mathrm{d}\boldsymbol{x} \, \mathrm{d}t.
\tag{A24}
$$

## Appendix B: The adjoint solution in the inverse and sensitivity problems

Assume that $(\boldsymbol{v}^i, q^i, \psi^i), i = 1, \ldots, d$, solves adjoint FS equations (13) in the steady state with observation of $u_i$ with $d = 2$ or 3

$$
\mathcal{F} = u_i(\boldsymbol{x}) = \int_\omega u_i\delta(\boldsymbol{x} - \bar{\boldsymbol{x}}) \, \mathrm{d}\bar{\boldsymbol{x}}, \; F_{\boldsymbol{u}} = \boldsymbol{e}^i\delta(\boldsymbol{x} - \bar{\boldsymbol{x}}), \; i = 1, \ldots, d,
\tag{B1}
$$

or observation of $h$ with $d = 1$

$$
\mathcal{F} = h(\boldsymbol{x}) = \int_\omega h\delta(\boldsymbol{x} - \bar{\boldsymbol{x}}) \, \mathrm{d}\bar{\boldsymbol{x}}, \; F_h = \delta(\boldsymbol{x} - \bar{\boldsymbol{x}}).
\tag{B2}
$$

Introduce the weight functions $w_i(\boldsymbol{x}), i = 1, \ldots, d$. It follows from (13) that $(w_i(\bar{\boldsymbol{x}})\boldsymbol{v}^i(\boldsymbol{x}), w_i(\bar{\boldsymbol{x}})q^i(\boldsymbol{x}), w_i(\bar{\boldsymbol{x}})\psi^i(\boldsymbol{x}))$ is a solution with $F_{\boldsymbol{u}} = w_i(\bar{\boldsymbol{x}})\boldsymbol{e}^i\delta(\boldsymbol{x} - \bar{\boldsymbol{x}})$ or $F_h = w(\bar{\boldsymbol{x}})\delta(\boldsymbol{x} - \bar{\boldsymbol{x}})$. Therefore, also

$$
\left( \int_\omega w_i(\bar{\boldsymbol{x}})\boldsymbol{v}^i \, \mathrm{d}\bar{\boldsymbol{x}}, \int_\omega w_i(\bar{\boldsymbol{x}})q^i \, \mathrm{d}\bar{\boldsymbol{x}}, \int_\omega w_i(\bar{\boldsymbol{x}})\psi^i \, \mathrm{d}\bar{\boldsymbol{x}} \right)
\tag{B3}
$$

is a solution with $F_{\boldsymbol{u}} = \int_\omega w_i(\bar{\boldsymbol{x}})\boldsymbol{e}^i\delta(\boldsymbol{x} - \bar{\boldsymbol{x}}) \, \mathrm{d}\bar{\boldsymbol{x}} = w_i(\boldsymbol{x})\boldsymbol{e}^i$ or $F_h = \int_\omega w(\bar{\boldsymbol{x}})\delta(\boldsymbol{x} - \bar{\boldsymbol{x}}) \, \mathrm{d}\bar{\boldsymbol{x}} = w(\boldsymbol{x})$. A sum over $i, i = 1, \ldots, d$, of each integral in (B3) is also a solution.





Consider a target functional $\mathcal{F}$ for the steady state solution with a weight vector $\boldsymbol{w}(\bar{\boldsymbol{x}})$ with components $w_i(\bar{\boldsymbol{x}})$ multiplying $\delta u^i$ in the first variation of $\mathcal{F}$. Using (16), $\delta\mathcal{F}$ is

$$
\begin{aligned}
\delta\mathcal{F} \;&= \int_\omega \boldsymbol{w}(\bar{\boldsymbol{x}}) \cdot \delta\boldsymbol{u}\,\mathrm{d}\bar{\boldsymbol{x}} = \int_\omega \sum_{i=1}^d w_i(\bar{\boldsymbol{x}})\delta u^i \,\mathrm{d}\bar{\boldsymbol{x}} \\
&= \int_\omega \sum_{i=1}^d w_i(\bar{\boldsymbol{x}}) \int_{\Gamma_b} f(\mathbf{T}\boldsymbol{u})\mathbf{T}\boldsymbol{u} \cdot \mathbf{T}\boldsymbol{v}^i \, \delta C \,\mathrm{d}\boldsymbol{x}\,\mathrm{d}\bar{\boldsymbol{x}} \\
&= \int_{\Gamma_b} f(\mathbf{T}\boldsymbol{u})\mathbf{T}\boldsymbol{u} \cdot \mathbf{T}\left( \int_\omega \sum_{i=1}^d w_i(\bar{\boldsymbol{x}})\boldsymbol{v}^i \,\mathrm{d}\bar{\boldsymbol{x}} \right) \delta C \,\mathrm{d}\boldsymbol{x}.
\end{aligned}
\tag{B4}
$$

**Appendix C: Steady state solution of the adjoint height equation in the FS model**

In a two dimensional vertical ice with $\boldsymbol{u}(x,z) = (u_1, u_3)^T$, the stationary equation for $\psi$ in (13) is

$$
(u_1\psi)_x = F_h + (\boldsymbol{h}\psi + F_{\boldsymbol{u}}) \cdot \boldsymbol{u}_z, \; z = h, \; 0 \le x \le L.
\tag{C1}
$$

When $x > x_*$, where $F_h = 0$ and $F_{\boldsymbol{u}} = 0$, we have $\psi(x) = 0$ since the right boundary condition is $\psi(L) = 0$.
If $u_1$ is observed at $\Gamma_s$ then $F(\boldsymbol{u}, h) = u_1(x)\chi(x)$ and $F_{\boldsymbol{u}} = (\chi(x), 0)^T$ and $F_h = 0$. The weight $\chi$ on $u_1$ may be a Dirac delta, a Gaussian, or a constant in a limited interval. On the other hand, if $F(\boldsymbol{u}, h) = h(x)\chi(x)$ then $F_h = \chi(x)$ and $F_{\boldsymbol{u}} = \boldsymbol{0}$.

Let $g(x) = u_{1z}(x)$ when $F_{\boldsymbol{u}} \ne \boldsymbol{0}$ and let $g(x) = 1$ when $F_h \ne 0$. Then by (24)

$$
(u_1\psi)_x - \boldsymbol{h} \cdot \boldsymbol{u}_z \psi = g(x)\chi(x).
\tag{C2}
$$

The solution to (C2) is

$$
\begin{aligned}
\psi(x) \;&= -\frac{1}{u_1(x)} \int_x^{x_*} \exp\left( -\int_x^\xi \frac{\boldsymbol{h} \cdot \boldsymbol{u}_z(y)}{u_1(y)} \,\mathrm{d}y \right) g(\xi)\chi(\xi)\,\mathrm{d}\xi, \; 0 \le x < x_*, \\
\psi(x) \;&= 0, \; x_* < x \le L.
\end{aligned}
\tag{C3}
$$

In particular, if $\chi(x) = \delta(x - x_*)$ then $\mathcal{F} = u_1(x_*)$ or $\mathcal{F} = h(x_*)$ and the multiplier is

$$
\psi(x) = -\frac{g(x_*)}{u_1(x)} \exp\left( -\int_x^{x_*} \frac{\boldsymbol{h} \cdot \boldsymbol{u}_z(y)}{u_1(y)} \,\mathrm{d}y \right), \; 0 \le x < x_*,
\tag{C4}
$$

which has a jump $-g(x_*)/u_1(x_*)$ at $x_*$.

**Appendix D: Simplified SSA equations**

The forward and adjoint SSA equations in (33) and (34) are solved analytically. The conclusion from the thickness equation in (33) is that

$u(x)H(x) = u(0)H(0) + ax = ax,$
\hfill (D1)



since $u(0) = 0$. Solve the second equation in (33) for $u$ on the bedrock with $x \leq x_{GL}$ and insert into (D1) using the assumptions for $x > 0$ that $b_x \ll H_x$ and $h_x \approx H_x$ to have

$$\frac{\rho g}{C} H^{m+1} H_x = \frac{\rho g}{C(m+2)} (H^{m+2})_x = -(ax)^m. \tag{D2}$$

The equation for $H^{m+2}$ for $x \leq x_{GL}$ is integrated from $x$ to $x_{GL}$ such that

$$
\begin{aligned}
H(x) &= \left( H_{GL}^{m+2} + \frac{m+2}{m+1} \frac{Ca^m}{\rho g} (x_{GL}^{m+1} - x^{m+1}) \right)^{\frac{1}{m+2}}, \\
u(x) &= \frac{ax}{H}, \quad H_x = -\frac{Ca^m}{\rho g} \frac{x^m}{H^{m+1}}.
\end{aligned}
\tag{D3}
$$

For the floating ice at $x > x_{GL}$, $\rho g H h_x = 0$ implying that $h_x = 0$ and $H_x = 0$. Hence, $H(x) = H_{GL}$. The velocity increases linearly beyond the grounding line

$$u(x) = ax/H(x) = ax/H_{GL}, \; x > x_{GL}. \tag{D4}$$

By including the viscosity term in (29) and assuming that $H(x)$ is linear in $x$, a more accurate formula is obtained for $u(x)$ on the floating ice in (6.77) of Greve and Blatter (2009).

**Appendix E:   Jumps in $\psi$ and $v$ in SSA**

Multiply the first equation in (34) by $H$ and the second equation by $u$ to eliminate $\psi_x$. We get

$$-Cmu^m v - \rho g H^2 v_x = HF_h - uF_u. \tag{E1}$$

Use the expression for $u$ and $H_x$ in (D3). Then

$$\rho g H(mH_x v - H v_x) = HF_h - uF_u, \tag{E2}$$

or equivalently

$$\left( \frac{v}{H^m} \right)_x = -\frac{1}{\rho g H^{m+2}} (HF_h - uF_u). \tag{E3}$$

The solutions $\psi(x)$ and $v(x)$ of the adjoint SSA equation (30) have jumps at the observation point $x_*$. For $x$ close to $x_*$ in a short interval $[x_*^-, x_*^+]$ with $x_*^- < x_* < x_*^+$, integrate (E3) to receive

$$\int_{x_*^-}^{x_*^+} \left( \frac{v}{H^m} \right)_x \, \mathrm{d}x = -\int_{x_*^-}^{x_*^+} \frac{HF_h - uF_u}{\rho g H^{m+2}} \, \mathrm{d}x. \tag{E4}$$

Since $H$ is continuous and $u$ and $v$ are bounded, when $x_*^- \to x_*^+$, then

$$v(x_*^+) - v(x_*^-) = -\frac{1}{\rho g H_*^2} \left( H_* \int_{x_*^-}^{x_*^+} F_h \, \mathrm{d}x - u_* \int_{x_*^-}^{x_*^+} F_u \, \mathrm{d}x \right). \tag{E5}$$





A similar relation for $\psi$ can be derived

$$\psi(x_*^+) - \psi(x_*^-) = \frac{1}{H_*} \int\limits_{x_*^-}^{x_*^+} F_u \, dx. \tag{E6}$$

With $F_u = 0$ and $F_h = 0$ for $x < x_*$ and $v(0) = \psi_x(0) = 0$, we find that

$$v(x) = \psi_x(x) = 0, \quad \psi(x) = \psi(x_*^-), \ 0 \leq x < x_*. \tag{E7}$$

5  If $F(u,h) = u\delta(x - x_*)$, then by (E5) and (E6)

$$v(x_*^+) = \frac{u_*}{\rho g H_*^2}, \quad \psi(x_*^+) - \psi(x_*^-) = \frac{1}{H_*}, \tag{E8}$$

and if $F(u,h) = h\delta(x - x_*)$, then

$$v(x_*^+) = -\frac{1}{\rho g H_*}, \quad , \psi(x_*^+) - \psi(x_*^-) = 0. \tag{E9}$$

## Appendix F: Analytical solutions in SSA

10  By Appendix E, $v(x) = 0$ for $0 \leq x < x_*$. Use equations in (34) with $H_x$ in (D3) for $x_* < x \leq x_{GL}$ to have

$$\frac{v_x}{v} = -\frac{axCmu^{m-1}}{\rho g H^3} = -\frac{Cmu^m}{\rho g H^2} = \frac{mH_x}{H}.$$

Let $\mathcal{H}(x - x_*) = \int_{-\infty}^{x-x_*} \delta(s) \, ds$ be the Heaviside step function at $x_*$. Then

$$v(x) = C_v H(x)^m \mathcal{H}(x - x_*), \quad 0 \leq x \leq x_{GL}. \tag{F1}$$

To satisfy the jump condition in (E8) and (E9), the constant $C_v$ is

$$C_v = \begin{cases} \dfrac{ax_*}{\rho g H_*^{m+3}}, & F(u,h) = u\delta(x - x_*), \\ -\dfrac{1}{\rho g H_*^{m+1}}, & F(u,h) = h\delta(x - x_*). \end{cases} \tag{F2}$$

Combine (F1) with the relation $\psi_x = (F_u - Cmu^{m-1}v)/H$ and integrate from $x$ to $x_{GL}$ to obtain

$$\psi(x) = C_v a^{m-1} C \left( x_{GL}^m - x^m \right), \ x_* < x \leq x_{GL}. \tag{F3}$$

With the jump condition in (E8) and (E9), $\psi(x)$ at $0 \leq x < x_*$ is

$$\psi(x) = \begin{cases} -\dfrac{1}{H_*} + \dfrac{Ca^m x_*}{\rho g H_*^{m+3}} \left( x_{GL}^m - x_*^m \right), & F(u,h) = u\delta(x - x_*), \\ -\dfrac{Ca^{m-1}}{\rho g H_*^{m+1}} \left( x_{GL}^m - x_*^m \right), & F(u,h) = h\delta(x - x_*). \end{cases} \tag{F4}$$




The weight for $\delta C$ in the functional $\delta\mathcal{L}$ in (31) is non-zero for $x_* < x \leq x_{GL}$

$$-vu^m = -C_v(ax)^m. \tag{F5}$$

Use (F1) and (34) in (31) to determine the weight for $\delta b$ in $\delta\mathcal{L}$,

$$\begin{aligned}
\psi_x u + v_x \eta u_x + v\rho g h_x &= \rho g(Hv)_x + F_h \\
&= C_v \rho g H^m \left[ (m+1)H_x \mathcal{H}(x - x_*) + H\delta(x - x_*) \right] + F_h.
\end{aligned} \tag{F6}$$

5  *Author contributions.* GC and PL designed the study. GC did the numerical computations. GC, PL and NK discussed and wrote the manuscript.

*Competing interests.* The authors declare that they have no conflict of interest.

*Acknowledgements.* This work was supported by Nina Kirchner's Formas grant 2017-00665. Lina von Sydow read a draft of the paper and helped us improve the presentation with her comments.



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
