# Peer review of "Sensitivity of ice sheet surface velocity and elevation to variations in basal friction and topography in the Full Stokes and Shallow Shelf Approximation frameworks using adjoint equations"

_The Cryosphere, 2020_

## Referee Comment (RC1) · Anonymous Referee #1 · 17 Jul 2020

**REFEREE REPORT FOR THE PAPER
"SENSITIVITY OF ICE SHEET SURFACE VELOCITY AND
ELEVATION TO VARIATIONS IN BASAL FRICTION AND
TOPOGRAPHY IN THE FULL STOKES AND SHALLOW SHELF
APPROXIMATION FRAMEWORKS" BY GONG CHENG, NINA
KIRCHNER, AND PER LOÖTSTEDT**

**Summary and High Level Discussion.** This paper is about using adjoint
calculus to determine the sensitivity of ice sheet surface velocities and elevation to
perturbations in basal friction and basal topogrophy. The ice sheet models are the
full Stokes and Shallow Shelf Approximation coupled with a time-dependent advection
equation for the kinematic free surface. The authors propose a few test cases with
both numerical and analytic solutions to the underlying forward and adjoint equations
and argue that it is necessary to include the time-dependent advection equation for
the ice surface elevation into the models. The reported findings show that: 1) there is a
delay in time between a perturbation at the ice base and the observation of the change
in elevation, 2) a perturbation at the base in the topography has a direct effect in
space at the surface above the perturbation and a perturbation in the basal friction is
propagated directly to the surface in time, and 3) perturbations with long wavelength
and low frequency will propagate to the surface while those of short wavelength and
high frequency are damped.

The topic of the paper is very interesting and it is worth publishing. However, it
needs a serious revision. The content as is presented is very difficult to digest. Below
I list several specific comments/recommendation:

**Comments.**
1. Introduction:
   (a) It is not entirely clear from the introduction (and abstract) what the
       motivation for running a sensitivity analysis is. It would be great if the
       authors could motivate this study and perhaps emhasize the impact of
       the sensitivity study results (in the intro especially) explicitely.
   (b) It would be beneficial to discuss the companion paper (Cheng and Lot-
       stedt, 2020) in more detail; in particular, what is the novelty in this
       paper compared to the previous one? If this companion paper would be
       useful for the reader to help him/her understand the (heavy) modeling
       part in this paper, it would be great to state this earlier or explicitly.
       Are some of the derivations in the Appendix also done in Cheng and
       Lotstedt, 2020? If so, perhaps the authors don't need to repeat these
       here.
   (c) In lines 18-19 on page 2, I would like to suggest the following reference
       for the inversion for the geothermal heat flux as well: Zhu, H., Petra,
       N., Stadler, G., Isaac, T., Hughes, T.J.R., Ghattas, O.: "Inversion of
       geothermal heat flux in a thermomechanically coupled nonlinear Stokes
       ice sheet model". The Cryosphere 10, 1477-1494 (2016).
2. How is $h(x,t)$ initialized, i.e., how is $h_0(x)$ defined?
3. Are there any constrints on $C$ in equation (4)? For instance, does it have
   to be positive? From line 13 it appears so. If this is the case, how are the
   authors making sure that this constant stays positive during inversion?

4. How are the Dirichlet boundary conditions set/defined, i.e., how are $u_u$ and $u_d$ set?

5. What is $H$ in equation 7? I assume this H is the height as shown in Figure 1a, please clarify.

6. In line 15, page 6, the authors state: "friction coefficient $C(x,t) \geq 0$, just as in the FS model. For the FS model it looks like $C > 0$, please clarify the possible equality here.

7. Second row, page 7: It is not clear how the adjoint equations have been derived. The authors say "Lagrangian of the forward equation?" (same in line 5, page 8). Do the authors mean the Lagrangian of the optimization problem governed by this PDE? What is the optimization objective function in the Appendix?

8. Line 9, pag 7: Need to define the topography $b(x)$.

9. Line 10, page 7: Please reformulate "its forward solution . . . ", it is not clear what solution we are talking about here. Same for the adjoint.

10. What do the authors mean by "The same forward and adjoint equations are solved both for the inverse problem and the sensitivity problem but with different forcing function $\mathcal{F}$", does this difference is due to inversion versus sensitivity or due to the fact the the objective is different for the two? In fact it is not clear how $\mathcal{F}$ is chosen for inversion versus sensitivity study. The authors gave a few examples for $\mathcal{F}$ but did not specify if $\mathcal{F}$ is or must be different. Same statement is made in line 5 on page 11 and similarly in line 5, page 25.

11. The last 2-3 lines on page 7 need to be explained more clearly. It sounds like there is an optimization/minimization problem solved, if so, what is the gradient? How is this optimization problem solved?

12. How is the nonlinear Stokes solved?

13. It would be beneficial to state the Lagrangian somewhere in the main text in order to help the reader follow the derivations and given expressions. This seems to be given in A15 for the Full Stokes, perhaps this should be moved to the main text.

14. Line 16, page 9: Why do the authors consider $e^i$?

15. The effect of the perturbations seems to be local. How do the authors choose where to induce these perturbations?

16. In general, it is difficult to follow all the variables, it would be great if the authors would remind the reader what is what. For instance I am not sure what the "perturbation $\delta u_1$" is (in the discussion for Fig 2 on page 10), is $u_1$ perturbed, or is it the effect of the perturbation in $C$ or basal friction on the velocity component $u_1$?

17. Please define exactly what "variation $\delta \mathcal{F}$ of the inverse problem" means? Similarly, what does the "variation of a functional" mean (e.g., in line 3, pag. 12))? Are these directional derivatives? It would be beneficial to show the mathematical definition in general and then apply it.

18. It is not clear how equations 22 and 23 are related.

19. Line 9, pag 18: What do the authors mean by "The relation in (38) . . . can also be interpreted as a way to quantify the uncertainty in $u$"? Please be more precise and define mathematically what you mean by "uncertainty". Same discussion needs more details in line 6 on page 25 and also in lines 5-6, page 3.

20. In general, this paper is difficult to follow. Perhaps the authors can add some roadmap to the begining of each section to guide the reader a bit through the research and findings. For instance I had to write out the sections to see how everything fits together because it got a bit impossible to navigate through so many setups and subsections. The structure seems to be the following:

    ```
    1. Introduction
    2. Ice Models
       2.1 Full Stokes
       2.2. Shallow shelf approximation.
    3. Adjoint equations
    3.1. Adjoint equations based on the FS model
         3.1.1. Time-dependent perturbations
         3.1.2. The sensitivity problem and the inverse problem.
         3.1.3. Steady state solution to the adjoint elevation equation in two dimensions.
    3.2. Shallow shelf approximation
         3.2.1. SSA in two dimensions.
         3.2.2. The two-dimensional forward steady state solution.
         3.2.3. The two-dimensional adjoint steady state solution with $F_u \neq 0$.
         3.2.4. The two-dimensional adjoint steady state solution with $F_h \neq 0$.
         3.2.5. The two-dimensional time dependent adjoint solution.
    ```

    (a) Sometimes the titles are not very representative or consistent, for instance Subsections 3.2.1 and 3.2.2 focus on forward equations and solutions eventhough Section 3.2. is called "Adjoint Equations", this is a bit confusing. Perhaps the authors should move forward problem matters to section 2.

    (b) Also, consider creating a table that summarizes all the examples and cases, shows the similarities and differences, parameter values, etc. and then refer back to this table from the sections and text. It is difficult to see the big picture with all the small subsections and various proposed scenarious.

    (c) The description of adjoints and problem setups are mixed with results. I recommend separating these to the extent possible.

    (d) Finally, there are several modeling information and parameter values inserted in the text which makes the reading of the actual research study and findings difficult. A table that summarizes somehow all these values might help to ease the discussion.

21. Line 1, pag. 25: Not sure what the point of the sentence " . . . confirm the conclusions here and are in good agreement with the analytical solutions." is here. Please add more details to explain.

22. Finally, the authors talk about sensitivity analysis, however throughout the paper the authors compute the effect of some perturbation in the parameters on some quantity of interest. To do a proper sensitivity analysis (or derive the sensitivity equations) one should look at the (total) derivative of the objective with respect to the parameter (of interest). This will give the equations to compute the sensitivity of the forward solution with respect to the parameter (or in finite dimensions to all the parameter components), etc. The authors should define clearly at the begining what they mean by "sensitivities" and how are these computed.

---

## Referee Comment (RC2) · Anonymous Referee #2 · 22 Jul 2020

This is generally speaking a good paper and clearly in terms of the numerical aspects a highly accomplished work.

Largely I have a very positive view of the manuscript, but the manuscript is not particularly well written or structured. My main worry is that the authors appear to have forgot to start their work by reading previous papers on the subject. In fact many of the statements presented in the paper as new findings, are not. For example the last three sentences in the abstract could have been in a number of previous papers, and arguably really just reflect common knowledge. Although, the sentence 'There is a delay in time between a perturbation at the ice base and the observation of the change in elevation' is actually not quite correct. (The surface topography responds immediately, but obviously it takes finite time for a finite-sized surface bump to be formed at the surface.)

The study is essentially numerical in nature. Similarly to other such numerical studies, this approach cannot really give a proper overview over the transformation of bed properties to the surface. Inherently such studies will be limited to giving some (typical) examples and to provide a flavor of what can be expected. On the other hand, this numerical allows for all non-linearities and finite-amplitude effects to be considered. I suggest that the authors do some rewriting and focus on the real strength and the novelty of their work. Fundamentally this a methodology paper where new time-dependent adjoint capabilities are developed and tested. This represents important progress in the field and is definitely publishable and of interest to the TC community. However, this is not a new theoretical study of study of the 'Sensitivity of ice sheet surface velocity and elevation to variations in basal friction and topography in the Full Stokes and Shallow Shelf Approximation frameworks' as a reader might be lead to believe based on the title.

The paper should be refocused and shortened. For example the introduction is very general and does not give the reader a feel for what the paper is really about. The adjoint approach does not give the sensitivity of velocities, topography, etc to a basal perturbations, thatis it does not give the derivatives du/db where u are surface velocities and b basal topography. It gives the derivative dI/db where I is a (scalar) cost function. In this paper the I is referred to as the Lagrangian function and is, for example, defined as the integral over surface velocities multiplied by a delta function in time and space. This limitation is inherent in the methodology used. In fact the adjoint method can be thought of as a computationally efficient approach to calculate dI/db without having to calculate the sensitivities du/db.

Arguably this makes the approach use less suitable for providing general information

about du/db than a calculation/estimate of dI/db. I can't see that the authors obtain any general results on the bed-to-surface that expand over and above what we know already from papers such as Gudmundsson, 2008. This is not to say that the paper does provide many new and valuable insights. However statements such as 'Perturbations in the friction coefficient at the base observed in the surface velocity determined by SSA are damped inversely' are arguably less specific that some previously published results. And a further example 'proportional to the wave number and the frequency of the perturbations in (40) and (45), thus making very oscillatory perturbations in space and time difficult to register at the ice sheet surface.' is not a particularly precise or informative statement. If the authors want to make statements about bed-surface relationships, forward or inverse, then they should consider replicating some of the previous work first, and then maybe expand on particular aspects.

I feel the authors missed a few citations. For example:

Monnier, J. and des Boscs, P.-E.: Inference of the bottom properties in shallow ice approximation models, Inverse Probl., 33(11), 115001, doi:10.1088/1361-6420/aa7b92, 2017.

I doubt the solution to (33) is new. I believe the same idea, and almost identical solutions, have been published many time before. For example see Eq. (8) in Weertman, 1961, and Nye 1959.

In summary, this manuscript should be shortened considerably and should focus on the development and testing of new time-dependent adjoint capabilities. I think this may actually not be that difficult, and may ultimately make the paper more readable and focused.

Citations Nye, J. F.: The Motion of Ice Sheets and Glaciers, J. Glaciol., 3(26), 493–507, doi:10.3189/S002214300001724X, 1959.

Weertman, J.: Stability of ice-age ice sheets, J. Geophys. Res., 66(11), 3783–3792,

doi:10.1029/JZ066i011p03783, 1961.

Gudmundsson, G. H. and Raymond, M.: On the limit to resolution and information on basal properties obtainable from surface data on ice streams, Cryosph., 2(2), 167–178, doi:10.5194/tc-2-167-2008, 2008.

---

## Author Response (AR1)

**Response to the reviewer 1**

September 29, 2020

**Summary and High Level Discussion**

This paper is about using adjoint calculus to determine the sensitivity of ice sheet surface velocities and elevation to perturbations in basal friction and basal topography. The ice sheet models are the full Stokes and Shallow Shelf Approximation coupled with a time-dependent advection equation for the kinematic free surface. The authors propose a few test cases with both numerical and analytic solutions to the underlying forward and adjoint equations and argue that it is necessary to include the time-dependent advection equation for the ice surface elevation into the models. The reported findings show that: 1) there is a delay in time between a perturbation at the ice base and the observation of the change in elevation, 2) a perturbation at the base in the topography has a direct effect in space at the surface above the perturbation and a perturbation in the basal friction is propagated directly to the surface in time, and 3) perturbations with long wavelength and low frequency will propagate to the surface while those of short wavelength and high frequency are damped.

The topic of the paper is very interesting and it is worth publishing. However, it needs a serious revision. The content as is presented is very difficult to digest. Below I list several specific comments/recommendation.

**Comments**

Because we rearranged the structure of the manuscript, all the equation and section numbers below refer to the unrevised version of this paper: `https://tc.copernicus.org/preprints/tc-2020-108/tc-2020-108.pdf`.

1. Introduction

   (a) It is not entirely clear from the introduction (and abstract) what the motivation for running a sensitivity analysis is. It would be great if the authors could motivate this study and perhaps emphasize the impact of the sensitivity study results (in the intro especially) explicitly.

   **Response:** We have expanded on the motivation for the sensitivity analysis and the relation to the inverse problem in Introduction,

before and in Sect. 3.1 and in Sect. 3.1.2. For example, certain perturbations at the base induce small perturbations at the surface. Such basal perturbations will be difficult to detect from observations at the surface. Consequently, these perturbations will not appear as a result of an inverse optimization based on surface data.

(b) It would be beneficial to discuss the companion paper (Cheng and Lötstedt, 2020) in more detail; in particular, what is the novelty in this paper compared to the previous one? If this companion paper would be useful for the reader to help him/her understand the (heavy) modeling part in this paper, it would be great to state this earlier or explicitly. Are some of the derivations in the Appendix also done in Cheng and Lötstedt, 2020? If so, perhaps the authors don't need to repeat these here.

**Response:** A short review of Cheng and Lötstedt, 2020, is written in the end of Introduction. Only the results of the derivations in Appendix are given in the companion paper with a marginal overlap of contents. It is noted in the end of Section 3.1.3 that the computed adjoint $\psi$ is very small in Cheng and Lötstedt, 2020, as expected by theory in the section. References to the companion paper concerning the SSA and FS equations are made at Conclusions (i) and (v) and after Conclusion (vi) in Section 3.2.3. The conclusions are motivated by numerical computations in Cheng and Lötstedt, 2020 and agree with the conclusions in this paper where they are motivated by analysis.

(c) In lines 18-19 on page 2, I would like to suggest the following reference for the inversion for the geothermal heat flux as well: Zhu, H., Petra, N., Stadler, G., Isaac, T., Hughes, T.J.R., Ghattas, O.: "Inversion of geothermal heat flux in a thermomechanically coupled nonlinear Stokes ice sheet model". The Cryosphere 10, 1477-1494 (2016).

**Response:** Thanks for the reference. We refer to it now in Introduction.

2. How is $h(x, t)$ initialized, i.e., how is $h_0(x)$ defined?

**Response:** $h_0$ could be any smooth function such that $h_0 > b$. The inequality is added.

3. Are there any constraints on $C$ in equation (4)? For instance, does it have to be positive? From line 13 it appears so. If this is the case, how are the authors making sure that this constant stays positive during inversion?

**Response:** $C$ is non-negative (changed now). We are not solving the inversion problem but since it is an optimization problem one can add inequality constraints $C \geq 0$ to be satisfied at the optimum in an optimization algorithm. There are numerical techniques to do that. The perturbation $\delta C$ at the base has to be such that $C + \delta C \geq 0$.

4. How are the Dirichlet boundary conditions set/defined, i.e., how are $u_u$ and $u_d$ set?

   **Response:** We consider the general case with the assumption in the adjoint problems that the inflow velocity $u_u$ and the outflow velocity $u_d$ of the ice are known. If the domain is at the ice divide then $u_u = 0$ and $u_d$ is the velocity at the calving front. If the latter is unknown, it can be computed with a different boundary condition there e.g. like (9) and then use $u$ at $\Gamma_d$ in the adjoint equations.

5. What is H in equation 7? I assume this H is the height as shown in Figure 1a, please clarify.

   **Response:** It is defined in the beginning of Sect 2.1 but we repeat it after (7).

6. In line 15, page 6, the authors state: "friction coefficient $C(x,t) \geq 0$, just as in the FS model. For the FS model it looks like $C > 0$, please clarify the possible equality here.

   **Response:** We have corrected the FS coefficient to $C \geq 0$.

7. Second row, page 7: It is not clear how the adjoint equations have been derived. The authors say "Lagrangian of the forward equation?" (same in line 5, page 8). Do the authors mean the Lagrangian of the optimization problem governed by this PDE? What is the optimization objective function in the Appendix?

   **Response:** The inverse problem is an optimization problem to find the optimal $b$ and $C$. In the sensitivity problem, $\delta b$ and $\delta C$ are known and we want to know the effect of these perturbations at the surface. In the inverse problem, we determine $\delta b$ and $\delta C$ iteratively such that $u + \delta u \rightarrow u_{\text{obs}}$. The relation between the inverse and sensitivity problems is discussed in an extended Section 3.1.2. The Lagrangian $\mathcal{L}$ for the sensitivity problem is defined in the appendices for SSA in (A4) and for FS in (A15). It differs from the Lagrangian of the inverse optimization problem in $F(u,t)$ in (10) (see the comment in the beginning of Section 3.1.2 and Section 2.2.5 in Cheng and Lötstedt 2020). The adjoint equations are the same except for the forcing terms $F_u$ and $F_h$. A comment about this is added after (13). Examples of different $F$-functions are found after (11).

8. Line 9, page 7: Need to define the topography $b(x)$.

   **Response:** The correction has been made.

9. Line 10, page 7: Please reformulate "its forward solution . . . ", it is not clear what solution we are talking about here. Same for the adjoint.

   **Response:** 'its' is replaced by 'the'

10. What do the authors mean by "The same forward and adjoint equations are solved both for the inverse problem and the sensitivity problem but

with different forcing function $F$", does this difference is due to inversion versus sensitivity or due to the fact the the objective is different for the two? In fact it is not clear how $F$ is chosen for inversion versus sensitivity study. The authors gave a few examples for $F$ but did not specify if $F$ is or must be different. Same statement is made in line 5 on page 11 and similarly in line 5, page 25.

**Response:** The $F$ term in the Lagrangian is different in the sensitivity and the inverse problems but the terms multiplied by the Lagrange multipliers $\psi, \mathbf{v}$, and $q$ are the same. This issue is discussed also in Comment 7. A better explanation of the difference is found after (12) and in Section 3.1.2.

11. The last 2-3 lines on page 7 need to be explained more clearly. It sounds like there is an optimization/minimization problem solved, if so, what is the gradient? How is this optimization problem solved?

**Response:** The optimization problem needs repeated solutions of the forward and adjoint problems to iteratively reach the minimum. The estimate of the sensitivity of perturbations is achieved by one forward and one adjoint solution. This is now elaborated on before Section 3.1 and in Section 3.1.2.

12. How is the nonlinear Stokes solved?

**Response:** They are solved by Elmer/Ice as we have written in a paragraph after (18).

13. It would be beneficial to state the Lagrangian somewhere in the main text in order to help the reader follow the derivations and given expressions. This seems to be given in A15 for the Full Stokes, perhaps this should be moved to the main text.

**Response:** The Lagrangian of the FS model is now also defined in the main text.

14. Line 16, page 9: Why do the authors consider $e^i$?

**Response:** We use the same unit vectors in Section 3.1.2. For instance, if we are interested in $\delta u_i$, $i = 1, 2$, or 3 at $(x_*, t_*)$ then

$$F_{\mathbf{u}} = \mathbf{e}^i \delta(\mathbf{x} - \mathbf{x}_*)\delta(t - t_*).$$

In the examples, we take $\mathbf{e}^1$.

15. The effect of the perturbations seems to be local. How do the authors choose where to induce these perturbations?

**Response:** The perturbations $\delta b$ and $\delta C$ can be local in space and time or more extended in space and time. They can be located anywhere on the bedrock. The effect at the surface is determined e.g. by the the formula (16). Since it is a linearization in $b$ and $C$, it is important that $\delta b$ and $\delta C$

are small. A comment about that is inserted before Section 3.1.1. The perturbations at the surface are always registered in one point $(x_*, t_*)$ in space and time but it can be chosen in many other ways by changing $F(u, h)$.

16. In general, it is difficult to follow all the variables, it would be great if the authors would remind the reader what is what. For instance I am not sure what the 'perturbation $\delta u_1$' is (in the discussion for Fig 2 on page 10), is $u_1$ perturbed, or is it the effect of the perturbation in $C$ or basal friction on the velocity component $u_1$?

    **Response:** This is clarified now.

17. Please define exactly what "variation $\delta\mathcal{F}$ of the inverse problem" means? Similarly, what does the "variation of a functional" mean (e.g., in line 3, pag. 12))? Are these directional derivatives? It would be beneficial to show the mathematical definition in general and then apply it.

    **Response:** This is explained in words now in Section 3.1.2. $\delta\mathcal{F}$ tells what happens to $\mathcal{F}$ when $b$ and $C$ change to $b + \delta b$ and $C + \delta C$.

18. It is not clear how equations 22 and 23 are related.

    **Response:** The paragraph has been reorganized with additional text.

19. Line 9, page 18: What do the authors mean by "The relation in (38)...can also be interpreted as a way to quantify the uncertainty in $u$"? Please be more precise and define mathematically what you mean by "uncertainty". Same discussion needs more details in line 6 on page 25 and also in lines 5-6, page 3.

    **Response:** Examples of uncertainties are given on page 3. They can be known estimates of measurement errors in $b$ and $C$. New text to explain uncertainty and a book on uncertainty are included on page 9. 'uncertainties' $->$ 'perturbations' on page 25.

20. In general, this paper is difficult to follow. Perhaps the authors can add some roadmap to the beginning of each section to guide the reader a bit through the research and findings. For instance I had to write out the sections to see how everything fits together because it got a bit impossible to navigate through so many setups and subsections. The structure seems to be the following:

    ```
    1. Introduction
    2. Ice Models
    2.1 Full Stokes
    2.2. Shallow shelf approximation.
    3. Adjoint equations
    3.1. Adjoint equations based on the FS model
    3.1.1. Time-dependent perturbations
    ```

```
3.1.2. The sensitivity problem and the inverse problem.
3.1.3. Steady state solution to the adjoint elevation equation in two dimensions.
3.2. Shallow shelf approximation
3.2.1. SSA in two dimensions.
3.2.2. The two-dimensional forward steady state solution.
3.2.3. The two-dimensional adjoint steady state solution with $F_u \neq 0$.
3.2.4. The two-dimensional adjoint steady state solution with $F_h \neq 0$.
3.2.5. The two-dimensional time dependent adjoint solution.
```

(a) Sometimes the titles are not very representative or consistent, for instance Subsections 3.2.1 and 3.2.2 focus on forward equations and solutions even though Section 3.2. is called "Adjoint Equations", this is a bit confusing. Perhaps the authors should move forward problem matters to section 2.

**Response:** Thanks for the comments. We have moved the forward SSA solutions in 2D to Section 2 and introduced new subsections with the numerical examples in Section 3.

(b) Also, consider creating a table that summarizes all the examples and cases, shows the similarities and differences, parameter values, etc. and then refer back to this table from the sections and text. It is difficult to see the big picture with all the small subsections and various proposed scenarios.

**Response:** We have written more about the MISMIP example in new Section 2.2.2 and refer to that in new Sections 3.1.2, 3.2.3, and 3.2.6.

(c) The description of adjoints and problem setups are mixed with results. I recommend separating these to the extent possible.

**Response:** The separation between theory and examples is more explicit now in Sections 3.1 and 3.2 and a change of the order of the paragraphs.

(d) Finally, there are several modeling information and parameter values inserted in the text which makes the reading of the actual research study and findings difficult. A table that summarizes somehow all these values might help to ease the discussion.

**Response:** We have collected parameters in Table 1 together with the description of the MISMIP test case in new Section 2.2.2.

21. Line 1 page 25, not sure what the point of the sentence "...confirm the conclusions here and are in good agreement with the analytical solutions." is here. Please add more details to explain.

**Response:** There is a brief summary of the paper now in Introduction. Reference is made to it here and there in the text and in the first paragraph of Conclusions. See also the reply to Comment 1b.

22. Finally, the authors talk about sensitivity analysis, however throughout the paper the authors compute the effect of some perturbation in the parameters on some quantity of interest. To do a proper sensitivity analysis (or derive the sensitivity equations) one should look at the (total) derivative of the objective with respect to the parameter (of interest). This will give the equations to compute the sensitivity of the forward solution with respect to the parameter (or in finite dimensions to all the parameter components), etc. The authors should define clearly at the begining what they mean by "sensitivities" and how are these computed.

   **Response:** We mean pointwise sensitivity in space and time at the ice surface. This is how we have chosen $F$ and $\mathcal{F}$. There is great freedom to choose another type of sensitivity by changing $F$ in (10). This is remarked in Introduction and before Section 3.1 there are examples of different $F$. With our $F$, the adjoint solution yields the perturbation $\delta u$ at a point in space and time on the surface for any basal perturbation $\delta C$. Solving the forward equation twice with $C$ and $C + \delta C$ yields the perturbation $\delta u = u(C + \delta C) - u(C)$ everywhere on the surface for a particular perturbation $\delta C$ at the base.

**Response to the reviewer 2**

September 29, 2020

Because we rearranged the structure of the manuscript, all the equation and section numbers below refer to the unrevised version of this paper: `https://tc.copernicus.org/preprints/tc-2020-108/tc-2020-108.pdf`.

This is generally speaking a good paper and clearly in terms of the numerical aspects a highly accomplished work.

Largely I have a very positive view of the manuscript, but the manuscript is not particularly well written or structured. My main worry is that the authors appear to have forgot to start their work by reading previous papers on the subject. In fact many of the statements presented in the paper as new findings, are not. For example the last three sentences in the abstract could have been in a number of previous papers, and arguably really just reflect common knowledge. Although, the sentence 'There is a delay in time between a perturbation at the ice base and the observation of the change in elevation' is actually not quite correct. (The surface topography responds immediately, but obviously it takes finite time for a finite-sized surface bump to be formed at the surface.)

**Response:** We have included about 65 references related to the subject in the paper and read them all. They are referred to in Introduction and in the other sections. It is true that some of the conclusions are found in other papers (e.g. about the damping of high frequency perturbations) but the analytical derivations and the explicit expressions are not found elsewhere. The results are valid for an entire ice sheet with variable height, not only for an ice slab with frozen coefficients in the PDEs. The effect of time variable perturbations in FS (Section 3.1.1) and SSA (Section 3.2.5) is new. There is a delay (or phase shift) in time when the full effect of a perturbation of the topography is observed in the elevation of the surface, see Figs. 3 and 9d. This holds true for the friction in SSA too in Fig 9c. For an oscillatory perturbation as in Fig. 3, it is fair to call this a delay in time. The weights $w_C$ and $w_b$ are $\approx 0$ for $\delta C$ and $\delta b$ at $(x_*, t_*)$ in Figs 9c,d indicating that a sudden change is visible only later when $w_C \neq 0$ and $w_b \neq 0$. A perturbation in the summer is growing at the surface reaching a maximum in the fall. This is also illustrated in a new example in Sect. 3.1.1 with a step perturbation where the surface effect is gradually growing from zero (as the reviewer remarks) but there is a delay in time when the full effect is reached.

The study is essentially numerical in nature. Similarly to other such numerical studies, this approach cannot really give a proper overview over the

transformation of bed properties to the surface. Inherently such studies will be limited to giving some (typical) examples and to provide a flavor of what can be expected. On the other hand, this numerical allows for all non-linearities and finite-amplitude effects to be considered. I suggest that the authors do some rewriting and focus on the real strength and the novelty of their work. Fundamentally this a methodology paper where new time-dependent adjoint capabilities are developed and tested. This represents important progress in the field and is definitely publishable and of interest to the TC community. However, this is not a new theoretical study of study of the 'Sensitivity of ice sheet surface velocity and elevation to variations in basal friction and topography in the Full Stokes and Shallow Shelf Approximation frameworks' as a reader might be lead to believe based on the title.

**Response:** With analysis of the adjoint FS equations in Section 3.1.1, we derive expressions for the influence of a time dependent and a time-independent perturbation $\delta C$ on a velocity component and the elevation in (17) and (18). Explicit expressions for how $\delta u$ and $\delta h$ depend on $\delta C$ and $\delta b$ in the SSA model are given in (38) and (42). We believe that these results are new. The advantage with analytical results compared to numerical results is that the dependence on the forward solution (e.g. $u$ and $H$) and parameters (e.g. $a$ and $C$) is apparent as the reviewer remarks. This is not the case with numerical calculations. The expressions are compared to time dependent and steady state numerical computations with FS and SSA in the companion paper Cheng and Lötstedt 2020. The differences in the forward solutions with and without perturbations, e.g. $\delta u = u(C + \delta C) - u(C)$, agree very well with the predictions using the adjoint techniques and the explicit formulas. The results from Cheng and Lötstedt 2020 are discussed now in several places in the text as a response to Referee 1. We view the investigation of the sensitivity as the main contribution and the adjoint equations as a tool to achieve that. The title reflects this in a better way now.

The paper should be refocused and shortened. For example the introduction is very general and does not give the reader a feel for what the paper is really about. The adjoint approach does not give the sensitivity of velocities, topography, etc to a basal perturbations, that is it does not give the derivatives du/db where u are surface velocities and b basal topography. It gives the derivative dI/db where I is a (scalar) cost function. In this paper the I is referred to as the Lagrangian function and is, for example, defined as the integral over surface velocities multiplied by a delta function in time and space. This limitation is inherent in the methodology used. In fact the adjoint method can be thought of as a computationally efficient approach to calculate dI/db without having to calculate the sensitivities du/db. Arguably this makes the approach use less suitable for providing general information about du/db than a calculation/estimate of dI/db.

**Response:** By choosing $F$ as in (12), the scalar functional $\mathcal{F}$ is $u_1(x, t)$, the $x$ component of the velocity. If we are interested in the $y$ component $u_2$ then $F$ will be slightly modified ($\mathbf{e}^1 \to \mathbf{e}^2$). The same is true for the $z$ component $u_3$. See also the Comment 14 of Reviewer 1. Later after (17), $F$ is chosen such that $\mathcal{F}$ is $h(x, t)$. Both $u_1, u_2, u_3$, and $h$ and the corresponding $\mathcal{F}$ are scalar variables.

In SSA in 2D in Section 3.2.1, the velocity $u$ is scalar. Examples of other $F$ (e.g. for the inverse problem) are found in the beginning of Section 3. Suppose that $u_1$ is observed at discrete $x_i$ and $C$ is perturbed at discrete $y_j$. Then the relation between $\delta u_1$ and $\delta C$ is $\delta u_{1i} = \sum_i W_{Cij}\delta C_j$ where $\mathbf{W}_C$ is a Jacobian matrix with elements $\partial u_{1i}/\partial C_j$ and can be determined by the adjoint approach. This is discussed in a new paragraph in Section 3.2.3 and a reference to Cheng and Lötstedt 2020 is made. For $\delta u_2$, $\mathbf{W}_C$ will be different. The relation between the sensitivity problem and the inverse problem is established in Section 3.1.2. See also the reply to Reviewer 1, Comments 1a, 7, and 22.

I can't see that the authors obtain any general results on the bed-to-surface that expand over and above what we know already from papers such as Gudmundsson, 2008. This is not to say that the paper does provide many new and valuable insights. However statements such as 'Perturbations in the friction coefficient at the base observed in the surface velocity determined by SSA are damped inversely' are arguably less specific that some previously published results. And a further example 'proportional to the wave number and the frequency of the perturbations in (40) and (45), thus making very oscillatory perturbations in space and time difficult to register at the ice sheet surface.' is not a particularly precise or informative statement. If the authors want to make statements about bed-surface relationships, forward or inverse, then they should consider replicating some of the previous work first, and then maybe expand on particular aspects.

**Response:** We have explicit expressions for the dependence of $\delta u$ and $\delta h$ on time independent parameters in SSA in (38) and (42). Using these formulas we derive an explicit expression (40) for how $\delta u$ depends on the wave number $k$ with much more detail than previously. As an example, the sensitivity to oscillatory perturbations increases as $1/H^4(x)$ when the ice is getting thinner closer to the GL and $x \to x_{GL}$. The expression is similar for $\delta h$, see the end of Section 3.2.4. The formulas (38) and (42) are valid for any type of perturbation, not just oscillatory ones. They tell how a basal perturbation at any $x$ is propagated to a perturbation at $x_*$ at the surface. The time dependent weight function is approximated in (45) for an expression for perturbations oscillating in time. The sentences quoted by the reviewer are summaries in words of the precise formula (40) and the approximation (45). A comparison is made with Gudmundsson's results in Section 3.2.2. In Gudmundsson 2008, FS for an ice slab is linearized with frozen coefficients and $n = m = 1$. Using Fourier analysis as in Gudmundsson 2008, it is necessary to have constant coefficients in the PDEs in the analysis and the perturbation at a point $x_*$ does not follow from the analysis. The coefficients depending on $u$ and $H$ are not frozen in our adjoint PDE (34) but vary with $x$ as they do in an ice sheet. In our formula (40), the expression in the parenthesis varies with $x_*$. Depending on $k$ and $x_*/x_{GL}$ the first two terms may cancel each other and then $\delta u_* \sim 1/k^2$. Our results are for the nonlinear SSA model with any $m > 0$ and, as a result of the weak influence of the friction, independent of $n$.

I feel the authors missed a few citations. For example: Monnier, J. and des Boscs, P.-E.: Inference of the bottom properties in shallow ice approximation

models, Inverse Probl., 33(11), 115001, doi:10.1088/1361-6420/aa7b92, 2017.

**Response:** Thanks for the reference. We refer to this paper now in Introduction.

I doubt the solution to (33) is new. I believe the same idea, and almost identical solutions, have been published many time before. For example see Eq. (8) in Weertman, 1961, and Nye 1959.

**Response:** We do not claim that the solution (35) of (33) is new but we need $u$ and $H$ in (35) to derive the solutions (38) and (42) to the adjoint equations. Three new sentences after (35) discuss Nye's and Weertman's solutions in relation to (35).

In summary, this manuscript should be shortened considerably and should focus on the development and testing of new time-dependent adjoint capabilities. I think this may actually not be that difficult, and may ultimately make the paper more readable and focused.

**Response:** We have argued above that there are results in the paper that are new and unique (not only the time dependent adjoint equations) and would prefer to keep these in the revised version. Examples are the solutions to the FS equations in Sections 3.1.1 and 3.1.3 and the explicit SSA solutions in Sections 3.2.3 and 3.2.4 and the propagation of $\delta b$ and $\delta C$ to $\delta u$ and $\delta h$. The results are all related to the sensitivity at the surface due to time dependent and time independent perturbations at the base. The editor and the other referee suggested no radical shortening of the paper. Our revised version contains the same material (somewhat expanded as a response to the editor and the referees) as the original version.

[revised manuscript text omitted]

---

## Author Response (AR2)

**Letter to the Editor and Response to the reviewers**

November 27, 2020

Dear Editor,

Our manuscript with the title 'Sensitivity of ice sheet surface velocity and elevation to variations in basal friction and topography in the Full Stokes and Shallow Shelf Approximation frameworks using adjoint equations' submitted to The Cryosphere has been revised following the all suggestions and comments by the referees. In particular, the abstract has been changed and the Introduction has been expanded somewhat.

The second referee suggests that the title of the paper should be changed. We would prefer to keep the present title since we consider the sensitivity results as the most interesting ones to the glaciological community. The time-dependent adjoint equations are a tool to derive these results. If the Editor prefers the title proposed by the second referee, we are willing to change it.

We hope that after these revisions the paper can be accepted for publication. We wish to thank the referees for their work and their detailed reading of the manuscript which helped us improve the presentation.

Best regards,
Cheng Gong, Nina Kirchner, Per Lötstedt

**Report 1**

Thank you for addressing all my comments. The only remaining item that needs a bit more attention is question 19. Thank you for explaining the sources of uncertainties. However, not sure what the point of the authors is if these uncertainties and their affect on the solution(s) are not formally quantified.
**Response:** Eq (42) gives the relation how the uncertainty is propagated from the parameters $b$ and $C$ to the solution $u$, or mathematically speaking $\delta u$ is a function of $\delta b$ and $\delta C$. As long as this relation is known, it is straightforward to quantify the effect of parameter uncertainties to the solution. An example is added in the discussion after (42).

**Report 2**

I still feel the authors are presenting number of results as new, that really are not. Let me put this in a positive way. By repeating well-know aspect of glacier flow in, for example, the last three sentences of the abstract, they are underselling their work by suggesting they merely rediscovered old facts rather than stressing the methodological advances that this work. What they do provide is a general numerical framework for studying the relationship between bed and surface properties.

**Response:** We have changed the Abstract as suggested by the reviewer by stressing the new framework and the time dependent analysis:*"Here, we present a general numerical framework for studying the relationship between bed and surface properties of ice sheets and glacier. Specifically, we use an inverse modeling approach and the associated time-dependent adjoint equations, derived in the framework of a Full Stokes model and a Shallow Shelf/Shelfy Stream Approximation model, respectively, to determine the sensitivity of ice sheet surface velocities and elevation to time-dependent perturbations in basal friction and basal topography."* One sentence about the damping of high frequency perturbations has been removed.

The emphasis on the finite-time it takes for a perturbation to be expressed at the surface appears misplaced to me. There is a direct and instantaneous effect. This of course is well known and does not need to be stressed.

**Response:** This may be well-known from the physical perspective, but not completely clear mathematically or numerically. Both effects are present. A change in $C$ in time is directly visible in $u$ at the surface (Eq (22), Fig. 3, Fig. 9a) and there is a delay in time between the change in $C$ or $b$ and the observation of it in $h$ at the surface (Eq (23), Fig. 3, Figs. 9c,d). The time delay is well illustrated in particular in Fig. 3. This is mentioned in Conclusions. We believe that the physical process is well represented by the numerical model, which confirms what is expected from the physical perspective.

What the manuscript is missing is an introduction over previous work. Either such section must be included, or the authors change the focus of the work to the methodological advances they make. This focus of the current introduction on global warming and RCPs' etc, is too general and does not really introduce the subject of the paper.

**Response:** We concluded from the initial review that no major changes were requested to our Introduction, but have now added a few sentences to the part of the Introduction focusing on the inverse problem where we discuss other sensitivity studies briefly and also in the end. However, most of the comparison to other sensitivity results is contained in Section 3.2.2.

This is clearly a very good piece of science and it deserves to be published. The authors have mostly avoided/refused to address my previous statements and continue to sell this as a new study on the relationships between bed and surface. While this is partly true, the real strength of the work is the development of the numerical adjoint approach. I feel that a title such as 'A general numerical framework for studying the relationship between bed and surface properties on

glaciers', is much truer to the actual focus of this study.

**Response:** We have underlined the development of the general framework for studying relations between bed and surface properties by adding a sentence in the Abstract. We are hesitant to changing the title of the manuscript at this stage because firstly, it is rather uncommon to do this so late in the review process, and secondly, and also more importantly, because we believe that the title in its present form is accessible to a broader audience interested in glaciological problems. We welcome the editor's advice on the issue of changing the title of the manuscript, and will follow it.

The manuscript is somewhat unevenly written. Have the co-authors spent enough time in helping with the formulation? It feels the manuscript needs some fresh eyes to go over it in detail and streamline it.

**Response:** All authors have carefully drafted, edited, read and re-read the manuscript several times, responding and taking into account the reviewers' comments. We have aimed to keep the text as accessible as possible to a broad audience, and which may be (mis)interpreted as not being streamlined to a more narrow recipient group. We are aware that especially numerical experts may have appreciated a more specialised or technically streamlined narrative. For the latter, we are happy to refer e.g. to our companion paper in The Cryosphere 14, p 673.

- I feel the citation of SSA should be to the original work by Morland, or alternatively to the first paper by MacAyeal (this is cited.) No need to attribute this idea to any later authors that then used this idea in later studies.

  **Response:** Citations of SSA in Sect. 2.2 are removed except MacAyeal.

- SSA in two dimensions might be misunderstood. I think you are referring to a flowline situation, ie to one horizontal dimension.

  **Response:** Thanks. We change it to 'The flowline model of SSA'

- Not sure Weertman included longitudinal stresses in his 1961 solution as stated in the manuscript. Is this correct? Do check this, not sure it is.

  **Response:** Yes, Weertman introduces the longitudinal stress in his formulas.

- L14 p 8 :'calibrated' $\longrightarrow$ scaled or normalized

  **Response:** Change to 'normalized'.

---

## Author Response (AR3)

**Reply to the Editor**

December 15, 2020

Dear Editor,

Our manuscript with the title 'Sensitivity of ice sheet surface velocity and elevation to variations in basal friction and topography in the Full Stokes and Shallow Shelf Approximation frameworks using adjoint equations' submitted to The Cryosphere has been revised following the all suggestions and comments by you.

In particular, we state from the beginning that we are interested in the perturbations in the grounded ice due to the perturbations at its base. The forward equations for FS have been expanded and the criteria satisfied by the grounding line in FS and SSA are now given explicitly.

We hope that after these revisions the paper can be accepted for publication. We wish to thank you and the referees once more for your work which helped us improve the presentation.

Best regards,
Cheng Gong, Nina Kirchner, Per Lötstedt

**Comments by the editor**

After a careful read of the last version of your manuscript, based on the two last reviews, I think there is still room for improvements. I have one main remark (which was already a point in my first review) as well as few minors points that should be accounted for before the final decision for publication is taken.

The main point concern the treatment of the grounding line dynamics for the transient cases (both FS and SSA). Indeed, solving for the GL dynamics in FS would require to solve the evolution equation for the bottom boundary $z_b$, in a similar form of (4a) for h. But this equation is missing in (4)? And one should also take care to enforce that $h > z_b >= b$. For the SSA equations (8), equation (8a), using $h_t$, implicitly assume that all changes in H impacts the upper surface elevation h, which is certainly true for grounded ice (where $z_b = b$), but doesn't hold anymore for floating shelves (where $z_b > b$), and where flotation criteria gives $h$ and $z_b$ as a function of H. So I think that $h_t$ should write $H_t$ in (8a) and that the flotation criteria should be given to determine $h$ from $z_b$ in the

grounded and floating cases. On the same lines, how is determined $x_{GL}$ in the analytical solution (12)? The proposed solution depends on the value of $x_{GL}$ but nothing is said on how it is obtained.

**Response:** Here is our response to the major remarks by the editor:

1. We mention in the revised abstract and Introduction that we are interested in the perturbations in the grounded ice due to perturbations at the base. Changes in $C$ and $b$ on the ground will have some effect on the floating ice but it will be less than on the grounded part. The boundary conditions at the wetted boundary of the floating ice are found in the new eq (5) and the complementarity condition defining the grounding line in FS is introduced. A new paragraph is written about the SSA equations and their grounding line in Sect. 2.2. We write in the beginning of Sect. 3 that we are interested in perturbations in the grounded part of the ice where $\delta C \neq 0$ and $\delta b$ directly affect the ice velocity and thickness. Thus, we can integrate over the ice from $b$ to $h$ in the $z$-direction.

2. We remark after former (12) (and present (13)) that $x_{GL}$ and $H(x_{GL})$ are assumed to be known. An alternative would be to use $H(0)$ at $x = 0$ as in Nye (1959) for the scaling (or boundary data) of the ice. The formula using $H(x_{GL})$ is more accurate compared to numerical solutions of FS and SSA. In the examples in Sect. 3.2.2, the solutions are perturbed around the computed steady state with computed $x_{GL}$ and $H(x_{GL})$.

3. There is a new paragraph in the end of Sect. 2.1 mentioning inequalities that the FS solution should fulfill, e.g. $H \geq 0$ and a bounded $\eta$. This should be guaranteed by the numerical method.

4. $h_t$ is replaced by $H_t$ in former (8) with a remark after the equations that we are interested in the grounded part where $h_t = H_t$.

**Minors remarks**

- page 4, line 12: not sure that ISCAL needs to be mentioned here as it is specific to the Elmer/Ice context

  **Response:** The sentence about ISCAL is expanded to explain the general applicability and where it is implemented.

- page 5, lie 17: In a two dimensional vertical ice $->$ For a two dimensional flow line geometry?

  **Response:** Changed.

- Eq. (2): I don't think that $\nu$ is used further in the manuscript so may be not necessary to define it?

  **Response:** It was also used in eq (6). But, no more than these two places. We decide to remove it.

- page 6, line 1: the equation for the friction law should be given

  **Response:** The expression of the friction law is added, the definition of the operator **T** is moved, accordingly.

- page 9, line 7: MISMIP is already used above. The acronym should be define at its first use.

  **Response:** Thanks. We have moved the definition to the first place it appears.

- Eq. (12): how do you explain that your analytical solution (starting from the same set of equations) is different than the one of Greve and Blatter (2019). Especially the uniform ice shelf thickness looks like a strange solution?

  **Response:** This is our approximated solution and the comparison to the solution in Greve and Blatter (2009) is mentioned above Fig. 2 and after eq (D4). We are only interested in the friction and the grounded ice, so this solution on the floating ice is just an approximation of the solution by Greve and Blatter. They assume that the thickness varies linearly, $H_x = const.$ while we let $H_x = 0$. We obtain $h_x = 0$ from the equations implying a flat upper surface. For a better approximation of the thickness on the floating ice, more work is needed.

- Eq. (14): should it be b or $z_b$ for the integral born?

  **Response:** Our focus is on the grounded ice where $C > 0$ and $z_b = b$. See also the response to the editor's major concerns. The floating ice equation for $z_b$ could have been included with a multiplier in (14) but the perturbations $\delta b$ and $\delta C$ are not meaningful under the floating ice. We could also have added an equation for the calving front in the functional. Taking into account the moving boundaries of the floating ice at the base and the front would probably need a separate paper.

- page 17, line 21: same questioning as above. You are mentioning a change in the bottom topography by delta b, whereas it should be delta $z_b$?

  **Response:** There is a discussion of this issue in the answer above. We consider $b$ and $C$ as external, given data which are perturbed.

- page 25, line 12: how the GL will move should be given in the initial set of equations.

  **Response:** How the grounding line is determined is found in new paragraphs in Sects. 2.1 and 2.2. We tell in a few words how this is done numerically in Sects. 3.1.2 and 3.2.4.